# NEED: Cross-Subject and Cross-Task Generalization for Video and Image Reconstruction from EEG Signals

**Shuai Huang**[*]
School of Optical-Electrical
and Computer Engineering
University of Shanghai
for Science and Technology
Shanghai, China
hhsavethworld@gmail.com

**Huan Luo**
School of Optical-Electrical
and Computer Engineering
University of Shanghai
for Science and Technology
Shanghai, China
luohuan2016@163.com

**Haodong Jing**
Institute of Artificial
Intelligence and Robotics
Xi'an Jiaotong University
Xi'an, China
Jinghd@stu.xjtu.edu.cn

**Qixian Zhang**
School of Computer
Science and Technology
Tongji University
Shanghai, China
zhangqx@tongji.edu.cn

**Litao Chang**
School of Optical-Electrical
and Computer Engineering
University of Shanghai
for Science and Technology
Shanghai, China
lilit0714@outlook.com

**Yating Feng**
School of Optical-Electrical
and Computer Engineering
University of Shanghai
for Science and Technology
Shanghai, China
243370889
@st.usst.edu.cn

**Xiao Lin**
School of Optical-Electrical
and Computer Engineering
University of Shanghai
for Science and Technology
Shanghai, China
242260497
@st.usst.edu.cn

**Chendong Qin**
School of Biomedical
Engineering
Shanghai Jiao Tong University
Shanghai, China
qcd2515427305@163.com

**Han Chen**
School of Optical-Electrical
and Computer Engineering
University of Shanghai
for Science and Technology
Shanghai, China
221240068
@st.usst.edu.cn

**Shuwen Jia**
School of Optical-Electrical
and Computer Engineering
University of Shanghai
for Science and Technology
Shanghai, China
231260077
@st.usst.edu.cn

**Siyi Sun**
Computer Science
University of Oxford
Oxford, United Kingdom
siyi.sun@cs.ox.ac.uk

**Yongxiong Wang**[†]
School of Optical-Electrical
and Computer Engineering
University of Shanghai
for Science and Technology
Shanghai, China
wyxiong@usst.edu.cn

---

[*]https://hharr1son.github.io/
[†]Corresponding author

39th Conference on Neural Information Processing Systems (NeurIPS 2025).

# Abstract

Translating brain activity into meaningful visual content has long been recognized as a fundamental challenge in neuroscience and brain-computer interface research. Recent advances in EEG-based neural decoding have shown promise, yet two critical limitations remain in this area: poor generalization across subjects and constraints to specific visual tasks. We introduce NEED, the first unified framework achieving zero-shot cross-subject and cross-task generalization for EEG-based visual reconstruction. Our approach addresses three fundamental challenges: (1) cross-subject variability through an Individual Adaptation Module pretrained on multiple EEG datasets to normalize subject-specific patterns, (2) limited spatial resolution and complex temporal dynamics via a dual-pathway architecture capturing both low-level visual dynamics and high-level semantics, and (3) task specificity constraints through a unified inference mechanism adaptable to different visual domains. For video reconstruction, NEED achieves better performance than existing methods. Importantly, Our model maintains 93.7% of within-subject classification performance and 92.4% of visual reconstruction quality when generalizing to unseen subjects, while achieving an SSIM of 0.352 when transferring directly to static image reconstruction without fine-tuning, demonstrating how neural decoding can move beyond subject and task boundaries toward truly generalizable brain-computer interfaces.

# 1 Introduction

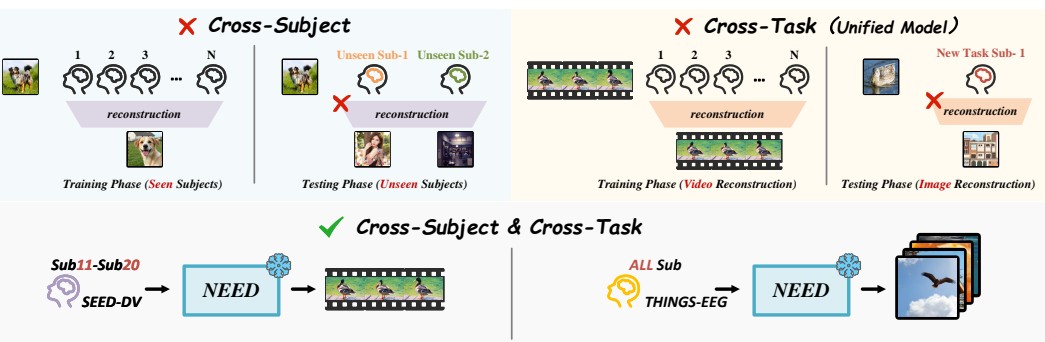

Figure 1: The dual challenge in neural decoding. Current models fail both to generalize to unseen subjects (left) and transfer between video and image reconstruction tasks (right). Our NEED framework addresses both challenges simultaneously.

Neural decoding for visual reconstruction from electroencephalography (EEG) signals represents a critical frontier at the intersection of neuroscience and artificial intelligence [1, 2]. This emerging technology offers profound implications across multiple domains: enabling new communication pathways for paralyzed individuals [3, 4], providing unprecedented insights into human visual perception [5, 6], advancing brain-computer interfaces [7, 8], and revealing fundamental cognitive mechanisms [9].

EEG signals offer distinct advantages for neural decoding despite their limited spatial resolution compared to other neuroimaging techniques [10, 11]. Their non-invasive nature, high temporal resolution, portability, and relatively low cost make EEG particularly suitable for real-world applications [12, 13]. The field of EEG-based visual decoding has advanced significantly in recent years, progressing from simple text reconstruction [14, 15, 16, 17] to increasingly complex visual content [18, 19]. Early work focused on classifying basic visual categories and reconstructing simple shapes [20, 21], while later research enabled reconstruction of static images with recognizable semantic content [22, 23, 24, 25]. Most recently, pioneering studies have demonstrated the possibility of reconstructing dynamic videos from EEG signals [26, 27, 28], representing a crucial step toward decoding the brain's perception of coherent scenes and motion [29, 30]. This progression toward decoding continuous visual stimuli

holds profound implications for understanding how the brain processes dynamic visual information and potentially recreating subjective perceptual experiences[31, 32].

Despite recent advances in reconstructing visual content from EEG signals, significant limitations persist that hinder practical applications, as illustrated in Fig. 1. **(1)Current neural decoders struggle with cross-subject generalization** [33, 34, 35, 36], demonstrating dramatic performance deterioration when models trained on specific individuals are applied to new subjects [37, 38]. This challenge arises from the inherently individualized nature of neural activity patterns, which vary due to anatomical differences, electrode positioning inconsistencies, and unique functional organizations of visual processing regions [39, 40]. The resulting models often capture subject-specific EEG patterns rather than generalizable neural representations of visual content [41]. **(2)Simultaneously, existing approaches remain constrained by task-specificity**[42], requiring dedicated frameworks for different stimulus types [43, 44, 45]. Models designed for static image reconstruction prove ineffective for dynamic video decoding without substantial architectural modifications and retraining, and the reverse limitation equally applies [46, 47]. This rigid task-dependency necessitates developing separate optimized frameworks for each visual decoding scenario, severely restricting scalability and real-world implementation. Cross-subject variability and task-specificity represent major challenges among several important bottlenecks in advancing neural decoding toward practical applications.

To address these limitations, we introduce *NEED* (**N**eural Decoding with **E**nhanced **E**xtensibility and **D**iversity), a unified framework for cross-subject and cross-task EEG-based visual reconstruction. NEED tackles those challenges in EEG decoding [48, 10, 49, 27, 50, 51, 35] through three key innovations: (1) an Individual Adaptation Module that normalizes subject-specific patterns, (2) a dual-pathway architecture capturing complementary visual information, and (3) a unified inference mechanism adaptable across tasks. Our experiments demonstrate NEED's ability to generalize to unseen subjects [52, 53, 36, 37] and transfer between video and image reconstruction tasks [42, 44, 45] without retraining—advancing neural decoding toward practical brain-computer interfaces that work across diverse individuals and visual experiences.

In conclusion, our contributions can be summarized as:

- We propose the first unified framework achieving zero-shot generalization across both EEG decoding tasks (image/video) and subjects.
- We develop DSGNet, a dual-stream EEG encoder capturing both spatial patterns and temporal dynamics of neural signals.
- We introduce a novel subject-adaptive module enabling robust multi-dataset knowledge transfer.
- We design a unified inference mechanism that flexibly integrates multiple guidance signals for adaptive reconstruction across visual domains.

## 2 Method

For more related work, see the Appendix A

### 2.1 Framework Overview

We introduce NEED, a unified neural decoding framework with zero-shot cross-subject and cross-task generalization capabilities for visual reconstruction from EEG signals. As illustrated in Fig. 2, NEED comprises three principal components: (1) Cross-Subject Adaptation with a pretrained Individual Adaptation Module (IAM) that addresses subject variability, (2) Dual-Pathway Understanding with parallel Perception and Semantic modules that overcome limited spatial resolution and complex temporal dynamics, and (3) a Unified Inference Mechanism that adaptively generates reconstructions across different tasks. The complete training and inference algorithm of our NEED framework is detailed in Algorithm in Appendix B.

### 2.2 Cross-Subject Adaptation

**Individual Adaptation Module (IAM)** To align heterogeneous EEG patterns across subjects, we propose the Individual Adaptation Module (IAM), a specialized architecture that transforms subject-specific signals into a normalized representation space. The core of IAM consists of two complemen-

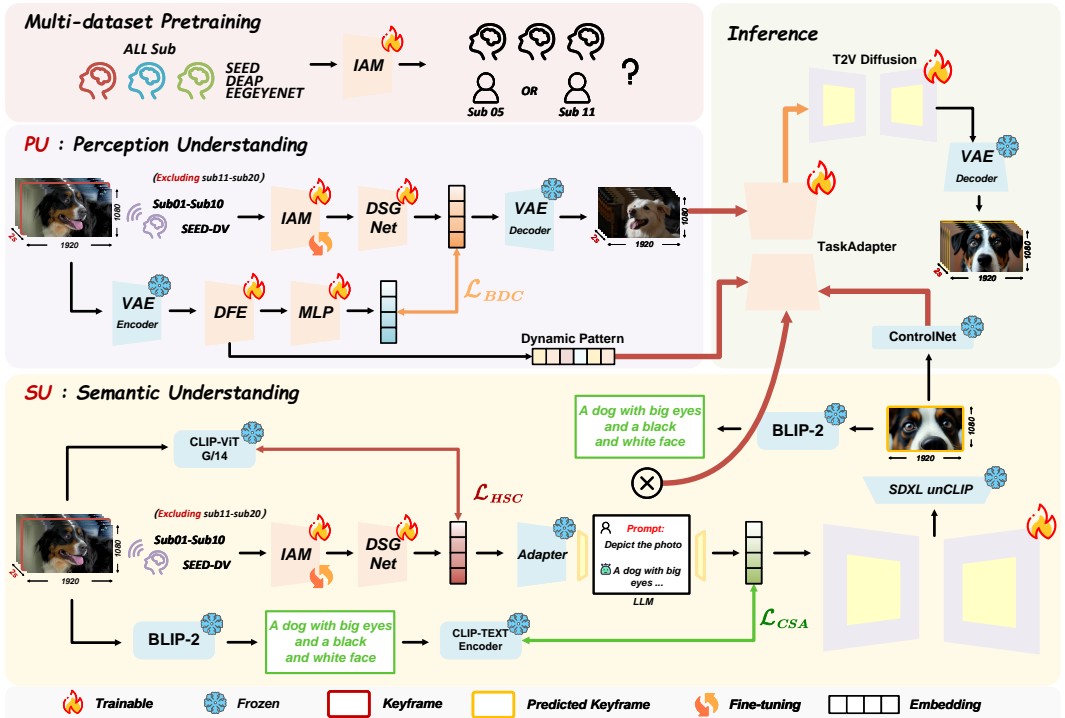

Figure 2: Overview of the NEED framework architecture. (1) Cross-Subject Adaptation with the Individual Adaptation Module (IAM) pretrained on multiple datasets to normalize subject-specific patterns; (2) Dual-Pathway Understanding with parallel Perception (PU) and Semantic (SU) modules that capture complementary visual information; and (3) a Unified Inference Mechanism that adaptively generates reconstructions across different tasks.

tary components: (1) a subject-aware attention mechanism that captures individual characteristics, and (2) a subject-invariant transformation that extracts shared neural patterns. Detailed architecture specifications and implementation details are provided in Appendix C.1.

For a given subject $i$, their EEG representation $E_i \in \mathbb{R}^{T \times d}$ is processed through a multi-head attention mechanism augmented by a learnable subject embedding $s_i \in \mathbb{R}^{d_s}$:

$$Z_{norm} = \underbrace{\sum_{h=1}^{H} W_O^h \left( \sigma \left( \frac{(E_i W_Q^h + s_i U_Q^h)(E_i W_K^h + s_i U_K^h)^T}{\sqrt{d_k}} \right) (E_i W_V^h) \right)}_{\text{Attention output } Z_{attn}} + \underbrace{\Gamma_\phi(E_i, \Omega_S, d_j)}_{\text{Style injection}} \quad (1)$$

where $\Gamma_\phi$ applies feature modulation guided by subject-agnostic knowledge $\Omega_S$ and dataset-specific embedding $d_j$ to accommodate different experimental paradigms. To enhance cross-subject generalization, IAM incorporates a hierarchical adaptation strategy that progressively normalizes signals from low-level characteristics to high-level patterns with $Z_{final} = \alpha_1 Z_{signal} + \alpha_2 Z_{spectral} + \alpha_3 Z_{temporal} + \alpha_4 Z_{norm}$, where coefficients are learned adaptively.

**Multi-Dataset Knowledge Transfer** IAM is pretrained on multiple diverse EEG datasets—SEED, DEAP, and EEGEYENET—to capture generalized neural response patterns across subjects with varying cognitive states. During pretraining, we optimize a multi-objective loss function $\mathcal{L}_{pretrain} = \alpha \mathcal{L}_{recon} + \beta \mathcal{L}_{adv}(\theta_D, \theta_G) + \gamma D_{KL}(p(Z_{norm}|S) \parallel p(Z_{norm}))$ where $\mathcal{L}_{recon}$ ensures reconstruction quality, $\mathcal{L}_{adv}$ implements adversarial subject disentanglement, and the KL divergence term minimizes subject-specific information in the representations.

### 2.3 Dual-Stream EEG Encoder

Our framework's foundation is a specialized dual-stream EEG encoder (***DSGNet***) that addresses limited spatial resolution and complex temporal dynamics in EEG signals through parallel processing

streams. This architecture efficiently extracts complementary information from both the topographical distribution and temporal evolution of neural activity. Detailed architecture specifications and implementation details are provided in Appendix C.2

**Spatial Stream Processing** The spatial stream leverages the inherent topographical structure of EEG recordings by representing electrode positions in a spherical coordinate system. We employ spherical harmonic embeddings to capture spatial patterns $E_{spatial}^{(0)} = \sum_{l=0}^{L} \sum_{m=-l}^{l} c_{l,m} Y_l^m(\theta_i, \phi_i)$ where $Y_l^m$ represent spherical harmonic functions and $c_{l,m}$ are learnable coefficients. These spatial embeddings are further refined through a graph convolutional network that models inter-electrode relationships via $E_{spatial}^{(l+1)} = \sigma(\tilde{D}^{-\frac{1}{2}} \tilde{A} \tilde{D}^{-\frac{1}{2}} E_{spatial}^{(l)} W^{(l)})$, where $\tilde{A} = A + I_N$ incorporates self-connections into the adjacency matrix.

**Temporal Stream Processing** The temporal stream captures dynamic neural responses across multiple time scales using a generalized Riemann-Liouville fractional transformation:

$$\mathcal{R}^\alpha(\mathbf{X})_t = \frac{1}{\Gamma(\alpha)} \int_0^t \frac{\mathbf{X}(s)}{(t-s)^{1-\alpha}} ds, \quad \alpha \in \{\alpha_1, ..., \alpha_m\} \tag{2}$$

This multi-scale approach with parameters $\alpha \in \{0.2, 0.4, 0.6, 0.8\}$ enables capturing both rapid neural responses and slower contextual dynamics. The transformed features are processed through dilated temporal convolutions with receptive field size $d^{(l)} = 2^{l-1}$ to model dependencies across various temporal ranges.

**Cross-Stream Integration** We implement a cross-stream attention mechanism that enables dynamic interaction between spatial and temporal representations $Z = \sigma(W_g \cdot [E_{spatial}; E_{temporal}; Z_{cross}] + b_g) \odot \tanh(W_f \cdot [E_{spatial}; E_{temporal}; Z_{cross}] + b_f)$ where $Z_{cross}$ represents cross-attention features computed via $Z_{cross} = \text{MultiHead}(Q = E_{spatial} W^Q, K = E_{temporal} W^K, V = E_{temporal} W^V)$. To enhance robustness against common EEG artifacts, we apply multi-dimensional masking $\tilde{E} = E \odot M_{spatial} \odot M_{temporal} \odot M_{frequency}$.

## 2.4 Dual-Pathway Understanding

**Perception Understanding Module** The Perception Understanding (PU) module extracts low-level visual dynamics essential for accurate reconstruction by integrating dual-stream encoder outputs with advanced temporal modeling. The module processes EEG signals through a cascaded architecture where $Z_p = \Psi_{PU}(\Phi_{enc}(\text{IAM}(E), \Theta_{spatio-temp}), \Theta_{task})$, with $\Phi_{enc}$ extracting spatial-temporal patterns from normalized EEG features, and $\Psi_{PU}$ providing task-adaptive processing. Our key innovation is the Dynamic Feature Extractor (DFE) with integrated Spatiotemporal Memory-Augmented Recurrent Network (SMARN) that decomposes visual dynamics through multi-level processing:

$$\begin{bmatrix} F_m \\ F_s \\ F_t \end{bmatrix} = \begin{bmatrix} \mathcal{F}_{motion}(\mathcal{T}_{3D}(V_0), \Theta_m) \\ \mathcal{F}_{spatial}(\mathcal{T}_{2D}(V_0), \Theta_s) \\ \text{SMARN}(\mathcal{T}_{TD}(V_0), H_{t-1}, \Theta_r) \end{bmatrix} \tag{3}$$

where $F_m$ encodes motion patterns, $F_s$ represents scene structures, and $F_t$ models temporal coherence through our novel SMARN architecture. $H_t = \sigma(W_h \cdot [V_t, H_{t-1}, M_t] + b_h)$ with $M_t = \sum_i \alpha_i M_{t-1}^i + \beta_t \cdot V_t$ formulates hierarchical memory representations with attentional weights $\alpha_i, \beta_t$ focusing on relevant temporal features. To establish effective correspondence between neural signals and visual dynamics, we employ a Bidirectional Dynamic Contrastive (BDC) alignment mechanism. Given EEG embeddings $Z_0$ and enhanced visual features $V_1$, we compute alignment using:

$$\mathcal{L}_{BDC} = -\sum_{i=1}^{N} \sum_{j=1}^{M} w_{ij} \log \frac{\exp(\text{sim}(Z_0^i, V_1^j)/\tau)}{\sum_k \exp(\text{sim}(Z_0^i, V_1^k)/\tau)} + \lambda \|\nabla_t Z_0 - \nabla_t V_1\|_2^2 \tag{4}$$

Here, $w_{ij}$ represents learnable attention weights for feature importance, $\text{sim}(\cdot, \cdot)$ is cosine similarity, $\tau$ controls distribution sharpness, and $\nabla_t$ denotes temporal gradients. This formulation ensures alignment in both feature space and temporal dynamics, enabling the model to identify neural patterns corresponding to specific visual elements while maintaining consistent temporal evolution.

**Semantic Understanding Module** The Semantic Understanding (SU) module operates in parallel with PU to extract high-level conceptual content. The module transforms EEG signals as $Z_s =$

$\Lambda_{SU}(\Xi_{adapt}(\text{IAM}(E), \Theta_{LM}), \Gamma_{context})$, where $\Xi_{adapt}$ bridges neural signals and semantic concepts via CLIP-ViT-G/14 keyframe features. Semantic alignment uses a Hierarchical Semantic Contrastive loss across abstraction levels:

$$\mathcal{L}_{HSC} = \sum_{l=1}^{L} \alpha_l \cdot \left( -\log \frac{\exp(Z_s^l \cdot C_k^l / \tau)}{\sum_n \exp(Z_s^l \cdot C_n^l / \tau)} \right) + \beta \cdot \|G(Z_s) - G(C_k)\|_F \tag{5}$$

where $L$ represents semantic levels from concrete to abstract, and $G(\cdot)$ computes the Gram matrix capturing semantic structure. We integrate language model capabilities via $Z_3 = \mathcal{F}_{LLM}(\mathcal{F}_{adapter}(Z_s))$ with Cross-modal Semantic Alignment $\mathcal{L}_{CSA} = \mathcal{L}_{cont}(Z_3, T) + \gamma \cdot \mathcal{L}_{KL}(Z_3, T) + \delta \cdot \mathcal{L}_{align}(Z_3, Keyframe)$ where $\mathcal{L}_{cont}$ is attention-weighted $w_{ij}$ contrastive learning between $Z_3$ and text embeddings $T$, $\mathcal{L}_{KL}$ is KL divergence between their distributions, and $\mathcal{L}_{align}$ ensures alignment with $BLIP2(Keyframe)$ embeddings. The enhanced embeddings $Z_4 = \mathcal{F}_{diffusion}(Z_3)$ condition SDXL unCLIP to generate predicted keyframes, enabling high-quality reconstruction across diverse visual tasks without task-specific training.

## 2.5 Unified Inference Mechanism

**Adaptive Task Conditioning** We implement an adaptive conditioning mechanism that modulates the reconstruction process based on the target task $Z_{task} = \text{TaskAdapter}([Z_p; Z_s], \tau_t)$, where $\tau_t$ indicates either video or image reconstruction. This mechanism automatically adjusts the processing pipeline based on the reconstruction target while maintaining a shared parameter space.

**Multi-Guidance Integration** The integrated conditioning is computed as: $c_t = \alpha_t \cdot \text{ControlNet}(\text{Keyframe}_{\text{pred}}) + \beta_t \cdot \text{TextEmbed}(\text{Caption}_{\text{fusion}}) + \gamma_t \cdot \text{Dyn}_{\text{pattern}} + \delta_t \cdot Z_{\text{eeg}}$. where $\text{Caption}_{fusion}$ is a fused textual description combining BLIP-2 generated captions from predicted keyframes and complementary semantic descriptions from EEG embeddings processed through a language model adapter. This dual-source caption fusion enhances semantic richness by incorporating both visual and neural information streams. The adaptive weights $[\alpha_t, \beta_t, \gamma_t, \delta_t] = \text{softmax}(\text{MLP}(z_t))$ dynamically adjust each guidance signal's contribution based on tasks.

**Unified Cross-Task Generation** Our framework achieves zero-shot cross-task generalization by leveraging shared neural patterns across visual modalities through a task-adaptive diffusion architecture. The diffusion process is defined as $x_T \sim \mathcal{N}(0, I)$, $x_{t-1} = \mu_\theta(x_t, t, c_t, \tau_t) + \sigma_t \cdot z$, where task encoding $\tau_t \in \{\tau_{video}, \tau_{image}\}$ explicitly differentiates between video and image processing, enabling a single model to adapt without structural modifications. For cross-task adaptation, we introduce conditional layer modulation $h_l(x_t, t, c_t, \tau_t) = \gamma_l(\tau_t) \odot \text{Norm}(h_{l-1}) + \beta_l(\tau_t)$, where $\gamma_l(\tau_t)$ and $\beta_l(\tau_t)$ dynamically transform feature spaces according to task requirements. This design creates a shared semantic representation space where $\mathcal{F}_{video}(E_{video}) \approx \mathcal{F}_{image}(E_{image})$. When processing image-related EEG signals, the model automatically configures the diffusion pathway by adjusting conditional signals while keeping all parameters $\theta$ fixed. Video reconstruction activates temporal attention modules with weight $\alpha_{temp} = \sigma(W_\tau \cdot \tau_{video})$, while image reconstruction shifts emphasis to spatial detail enhancement with $\alpha_{spat} = \sigma(W_\tau \cdot \tau_{image})$ while effectively zeroing temporal components $\alpha_{temp} \approx 0$. The model adapts feature representation distributions $p(Z|\tau_t)$, enabling conditional transformation $P(x_{t-1}|x_t, \tau_{video}) \neq P(x_{t-1}|x_t, \tau_{image})$ without architectural changes.

# 3 Experiments

## 3.1 Datasets

Our NEED framework leverages multiple EEG datasets to enable zero-shot cross-subject and cross-task generalization. We use SEED-DV [26] as our primary training dataset, containing EEG recordings from 20 subjects watching 1,400 video clips across 40 visual categories, recorded with a 62-channel cap at 1000 Hz. For IAM pretraining, we utilize SEED [54], DEAP [55], and EEGEYENET [56], providing diverse data for robust subject-invariant representations. For cross-task generalization evaluation, we test on THINGS-EEG [57], which contains recordings from 10 subjects viewing 22,000 static images across 1,854 object categories. Full dataset details and preprocessing are provided in Appendix D. .

## 3.2 Experimental Settings

We preprocess EEG data with standard filtering (0.5-40 Hz) and normalize sampling rates to 200 Hz across datasets. Our framework handles channel configuration differences across datasets through the IAM module, using a cross-dataset adaptation strategy detailed in Appendix C.3. Training proceeds in three stages: IAM pretraining on multiple datasets for subject-invariant representations, dual-pathway architecture training on SEED-DV, and unified inference mechanism fine-tuning. We evaluate using perceptual metrics (SSIM, PSNR), semantic metrics (CLIP-based classification), and temporal coherence measures (CLIP-pcc, FVD). Performance retention rate quantifies generalization capability relative to within-subject/within-task performance. Detailed implementation specifics are provided in Appendix E.

# 4 Results

## 4.1 Visual Perception Benchmark

We evaluate ***DSGNet***(the EEG encoder used in our NEED framework) on visual perception EEG data collected during subjects viewing various stimuli. Table 1 compares DSGNet against established EEG classifiers and recent deep learning approaches. DSGNet consistently outperforms all baselines across all tasks with statistical significance ($p < 0.05$). The improvement is particularly pronounced in challenging multi-class scenarios, with relative accuracy gains of 70.9% and 23.9% for 40-class and 9-class top-1 classification respectively. Even in binary tasks where conventional methods approach higher accuracy, DSGNet maintains 7-10% relative improvements.

Table 1: Average classification accuracy (%) and standard deviation across all subjects with different EEG classifiers on visual perception tasks. The star symbol (*) represents results above chance level with statistical significance (two-sample t-test: $p < 0.05$).

| Methods | | Multi-class Classification | | | | | Binary Classification | | | |
|---|---|---|---|---|---|---|---|---|---|---|
| | | 40-c top-1 | 40-c top-5 | 9-c top-1 | 9-c top-3 | Color | Fast/Slow | Numbers | Human Face | Human |
| Chance level | | 2.50 | 12.50 | 11.11 | 33.33 | 20.57 | 50.00 | 65.64 | 62.25 | 71.43 |
| **EEG Features** | | | | | | | | | | |
| SVM (PSD)[58] | | 5.19/2.81* | - | 19.02/3.27* | - | 21.31/2.97 | 53.56/1.11* | 64.15/1.22 | 58.94/2.21 | 70.91/1.84 |
| MLP (PSD) | | 6.20/3.02* | 18.91/5.94* | 21.59/3.00* | 49.86/3.78* | 22.02/3.27 | 55.15/1.20* | 64.48/0.92 | 63.94/1.13 | 71.74/1.76 |
| GLMNet (PSD)[26] | *[NIPS'24]* | 6.23/2.91* | 18.95/5.62* | 21.69/3.20* | 50.03/4.10* | 26.40/2.99* | 55.42/1.32* | 64.68/0.92 | 64.22/1.43 | 72.27/1.57 |
| SVM (DE)[58] | | 4.82/2.80* | - | 19.05/3.39* | - | 21.07/2.88 | 53.34/1.25* | 63.62/1.73 | 57.82/3.50 | 0.25/1.94 |
| MLP (DE) | | 6.12/3.08* | 19.02/5.71* | 21.17/3.24* | 49.40/4.94* | 25.91/3.27* | 54.76/1.25* | 64.10/0.70 | 63.41/1.57 | 71.74/1.76 |
| GLMNet (DE)[26] | *[NIPS'24]* | 6.16/3.18* | 19.12/6.07* | 21.34/3.34* | 49.55/4.57* | 26.15/3.24* | 55.06/1.20* | 64.25/0.74 | 63.63/1.80 | 72.27/1.58 |
| **Raw EEG Signals** | | | | | | | | | | |
| ShallowNet[6] | *[HBM'17]* | 5.59/2.27* | 16.93/4.66* | 21.40/1.96* | 49.62/2.34* | 27.00/2.09* | 56.62/1.77* | 66.15/0.89 | 64.87/1.54 | 73.21/1.52 |
| DeepNet[6] | *[HBM'17]* | 4.56/1.52* | 14.30/3.25* | 20.27/1.25* | 48.06/1.59* | 26.37/1.95* | 55.42/0.59* | 65.71/0.24 | 61.58/3.93 | 72.86/0.40 |
| EEGNet[10] | *[JNE'18]* | 4.64/0.86* | 14.25/1.87* | 19.63/0.81* | 47.04/1.45* | 25.46/1.31* | 51.99/2.00 | 64.67/0.60 | 61.37/1.31 | 72.38/0.98 |
| TSCNet[8] | *[SMC'20]* | 6.84/2.15* | 18.26/3.95* | 22.18/1.76* | 50.62/2.38* | 27.85/1.63* | 56.98/1.54* | 66.54/0.87 | 65.42/1.35 | 73.51/1.28 |
| Conformer[48] | *[TNSRE'22]* | 4.93/1.57* | 15.36/4.44* | 20.92/0.98* | 49.25/1.49* | 27.53/1.37* | 55.02/0.83* | 65.73/0.26 | 64.96/1.14 | 73.00/0.85 |
| BraVL[2] | *[TPAMI'23]* | 7.05/2.23* | 19.05/3.75* | 22.53/1.84* | 51.07/2.41* | 28.12/1.52* | 57.88/1.64* | 66.92/0.83 | 65.76/1.38 | 73.65/1.23 |
| EEGPT[13] | *[NIPS'24]* | 6.95/2.63* | 18.85/4.11* | 22.10/1.96* | 50.55/2.33* | 27.95/1.73* | 57.10/1.85* | 66.75/0.94 | 65.35/1.42 | 73.25/1.33 |
| TSConv[19] | *[ICLR'24]* | 4.92/0.99* | 15.05/2.31* | 20.00/1.01* | 47.76/1.51* | 26.89/1.83* | 55.32/0.99* | 65.39/0.41 | 64.39/1.47 | 72.68/0.67 |
| GLMNet[26] | *[NIPS'24]* | 6.20/3.02* | 17.75/4.24* | 21.93/1.87* | 50.01/2.52* | 27.33/1.45* | 57.35/1.98* | 66.21/0.91 | 65.10/1.45 | 73.34/1.31 |
| **DSGNet (Ours)** | *[This work]* | **14.23/3.12*** | **25.75/4.44*** | **29.93/2.07*** | **58.01/2.72*** | **35.33/1.65*** | **65.35/2.18*** | **74.21/1.11** | **73.10/1.65** | **81.34/1.51** |
| *Relative Improvement of DSGNet over best baseline (%)* | | | | | | | | | | |
| | | +101.8% | +35.2% | +32.8% | +13.6% | +25.6% | +12.9% | +10.9% | +11.2% | +10.4% |

## 4.2 Video Reconstruction Results

Table 2 presents our framework's performance across video and image reconstruction tasks compared to existing methods. For SEED-DV video reconstruction, our approach achieves superior semantic accuracy (0.898 in 2-way classification) while maintaining competitive visual fidelity (SSIM: 0.356). Ablation studies reveal the Individual Adaptation Module as most critical, with performance dropping dramatically without it (0.898→0.642 in 2-way accuracy). Both PU and SU pathways contribute complementarily to overall reconstruction quality. Fig. 3 shows cases representative reconstructions. Successful cases demonstrate our framework's ability to capture both semantic content and temporal dynamics across diverse scenes. Failed cases (red) reveal limitations with complex motion patterns and lighting conditions.

Table 2: Comparison of visual reconstruction performance across EEG, fMRI, and MEG-based methods for video and image reconstruction tasks. More detailed ablation experiments are shown in the Appendix F.1.1 and Appendix F.1.2

| Model Variant | Video Reconstruction | | | | | Image Reconstruction | | | | |
| | Video-based | | Frame-based | | | Low-level | | High-level | | |
| | 2-way↑ | 40-way↑ | 2-way↑ | 40-way↑ | SSIM↑ | PixCorr↑ | SSIM↑ | CLIP↑ | Inception↑ | SwAV↓ |
|---|---|---|---|---|---|---|---|---|---|---|
| EEG2video (EEG)[26] | 0.852±0.02 | 0.340±0.01 | 0.798±0.03 | 0.232±0.02 | 0.300±0.03 | - | - | - | - | - |
| Wang (fMRI)[47] | 0.773±0.03 | - | 0.713±0.04 | - | 0.118±0.08 | - | - | - | - | - |
| Kupershmidt (fMRI)[46] | 0.771±0.03 | - | 0.764±0.03 | 0.179±0.02 (50-way) | 0.135±0.08 | - | - | - | - | - |
| MinD-Video (fMRI)[30] | 0.839±0.03 | 0.197±0.02 (50-way) | 0.796±0.03 | 0.174±0.03 (50-way) | 0.171±0.08 | - | - | - | - | - |
| NeuroClips (fMRI)[45] | 0.834±0.03 | 0.220±0.01 (50-way) | 0.806±0.03 | 0.203±0.01 (50-way) | 0.390±0.08 | - | - | - | - | - |
| ATM-S (EEG)[45] | - | - | - | - | - | 0.155 | 0.330 | 0.786 | 0.730 | 0.582 |
| CognitionCapturer (EEG)[44] | - | - | - | - | - | 0.150 | 0.347 | 0.715 | 0.669 | 0.580 |
| Mind's Eye (fMRI)[25] | - | - | - | - | - | 0.295 | 0.317 | 0.918 | 0.929 | 0.382 |
| META-MEG (MEG)[40] | - | - | - | - | - | 0.072 | 0.320 | 0.683 | 0.702 | 0.615 |
| **Full Model (Ours) (EEG)** | **0.898±0.02** | **0.405±0.02** | **0.839±0.03** | **0.293±0.02** | 0.356±0.02 | 0.162 | **0.352** | 0.795 | 0.742 | 0.575 |
| *w/o IAM* | 0.642±0.03 | 0.104±0.02 | 0.587±0.04 | 0.085±0.02 | 0.171±0.02 | 0.054 | 0.063 | 0.421 | 0.398 | 0.732 |
| *w/o PU* | 0.723±0.02 | 0.293±0.02 | 0.678±0.04 | 0.184±0.03 | 0.201±0.03 | 0.092 | 0.116 | 0.563 | 0.526 | 0.694 |
| *w/o SU* | 0.804±0.03 | 0.146±0.03 | 0.756±0.03 | 0.102±0.02 | 0.278±0.02 | 0.127 | 0.153 | 0.654 | 0.613 | 0.621 |
| *w/o Task Cond.* | 0.845±0.03 | 0.336±0.02 | 0.786±0.02 | 0.228±0.03 | 0.295±0.04 | 0.043 | 0.063 | 0.389 | 0.352 | 0.793 |
| *w/o DynPattern* | 0.836±0.02 | 0.327±0.02 | 0.743±0.04 | 0.217±0.03 | 0.284±0.03 | 0.109 | 0.124 | 0.623 | 0.575 | 0.659 |
| *w/o ControlNet* | 0.825±0.03 | 0.312±0.03 | 0.767±0.02 | 0.193±0.02 | 0.263±0.03 | 0.123 | 0.152 | 0.642 | 0.592 | 0.625 |
| *w/o Caption* | 0.831±0.02 | 0.298±0.02 | 0.775±0.03 | 0.205±0.03 | 0.271±0.02 | 0.115 | 0.146 | 0.607 | 0.584 | 0.638 |

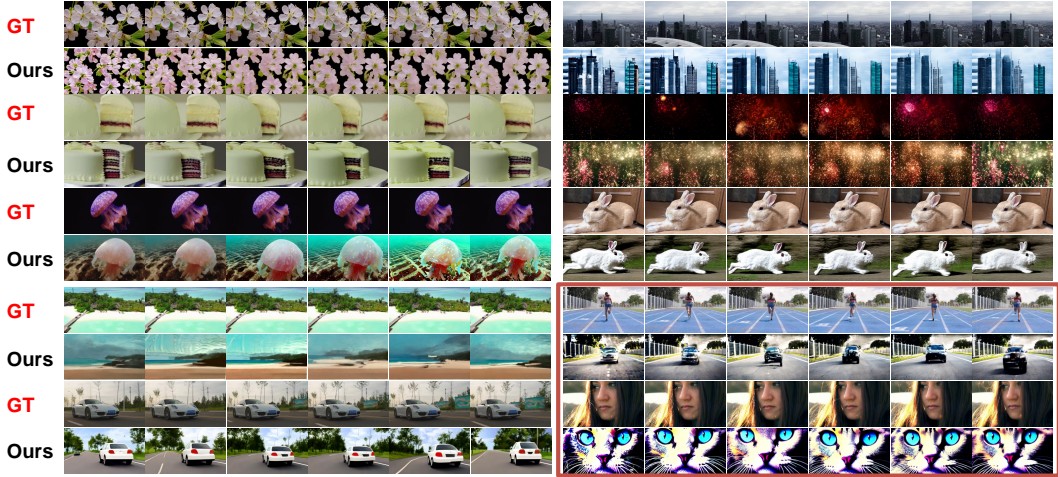

Figure 3: Sample video reconstruction results from our NEED framework on the SEED-DV dataset. Red: Failed cases.

## 4.3 Cross-Subject Generalization and Temporal Analysis

Fig 4 visualizes NEED's cross-subject generalization capabilities as training subject count increases. Classification accuracy (Figure 6a) shows consistent improvement across all metrics, with 2-way accuracy reaching 84.1% using 19 subjects. Quality metrics (Figure 6b) follow similar trends, with CLIP-pcc improving most substantially (0.128→0.682) while FVD decreases (584.7→197.5). Notable performance jumps occur between 5-10 and 10-15 subjects, suggesting threshold effects in capturing generalizable neural patterns. Fig 4c-d displays temporal analysis across subjects, revealing consistent information accumulation patterns despite individual variations. Further supporting analyses are available in Appendix F.2 and Appendix F.5. These results demonstrate our framework's robust ability to generalize across unseen subjects while preserving temporal dynamics.

## 4.4 Cross-Task Generalization

Table 2 presents our framework's zero-shot transfer capabilities on the THINGS-EEG static image reconstruction task. Without fine-tuning, NEED maintains competitive performance across all metrics compared to task-specific models, achieving the highest SSIM (0.352) among EEG-based methods. Task conditioning proves critical for enabling cross-task transfer, with performance dropping

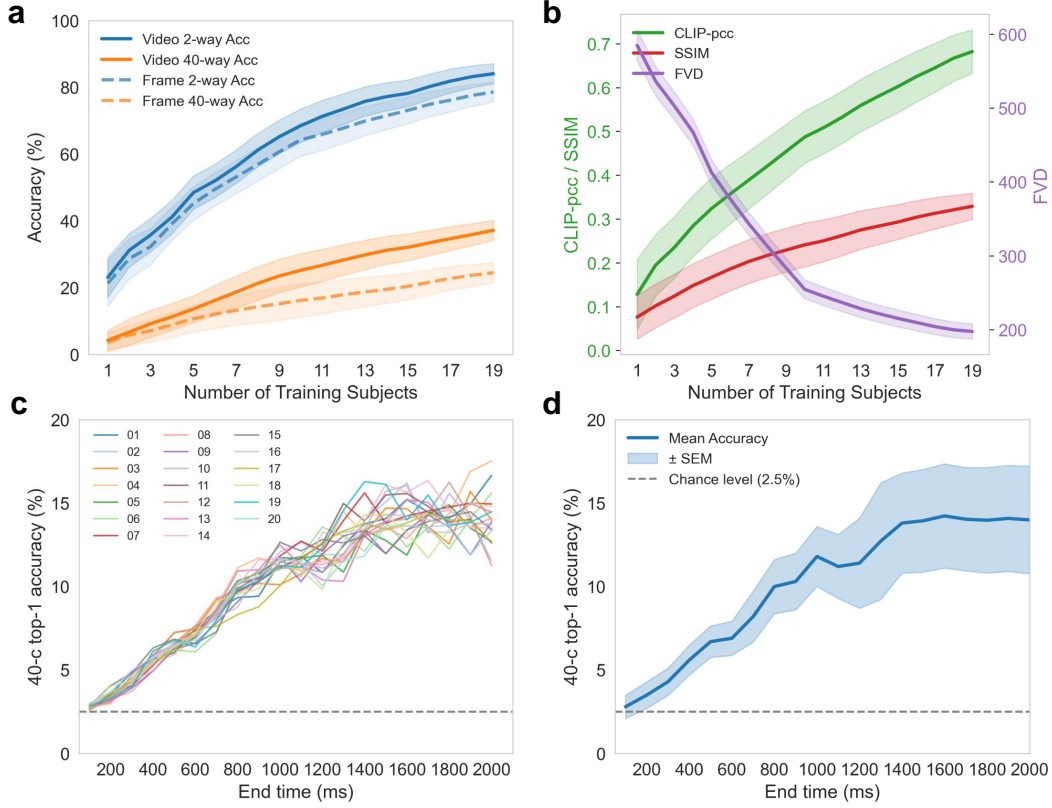

Figure 4: Cross-subject analysis on the SEED-DV dataset. (a) Classification accuracy vs. training subject count. (b) Reconstruction quality metrics vs. training subject count. (c) Individual 40-class top-1 accuracy curves for all subjects. (d) Average accuracy with standard error.

dramatically without this component (SSIM from 0.352 to 0.063). The PU and SU modules contribute complementarily to visual details and semantic accuracy, demonstrating our architecture's effective handling of both dynamic videos and static images through task-adaptive mechanisms.

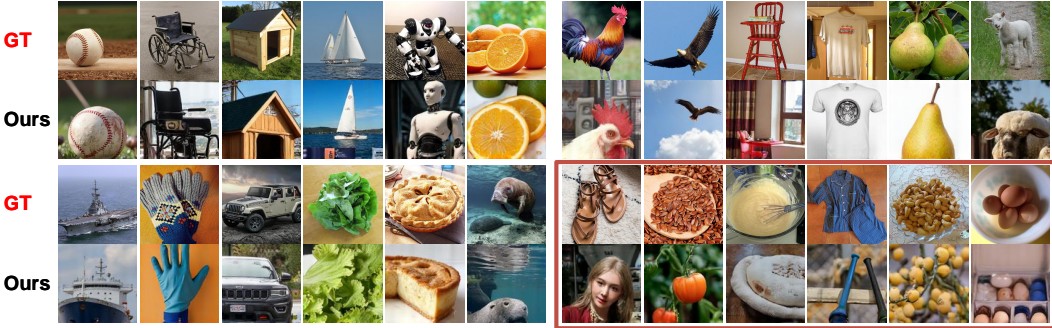

Figure 5: Zero-shot cross-task generalization results for static image reconstruction on the THINGS-EEG dataset. Red: Failed cases.

Fig. 5 shows representative examples of zero-shot image reconstruction. Successful cases demonstrate our framework's ability to capture key visual features and semantic content despite training solely on video data. Failed cases (red) reveal current limitations in reconstructing fine details and complex object arrangements when transferring to static image tasks.

## 4.5 Brain Region Contribution Analysis

Different brain regions exhibit distinct contributions to neural decoding. Fig 6a shows classification accuracy when ablating specific regions, with occipital areas showing lowest performance individually (8.1%) while full-brain integration yields highest accuracy (14.3%). For video reconstruction (Fig 6b), occipital-only signals capture basic scene structure but miss object details, while adding temporal regions improves semantic content. Dynamic scenes require parietal involvement for effective motion representation. Fig 6c illustrates the electrode distribution across five brain regions used in our analysis. Similarly, static image reconstruction (Fig 6d) demonstrates quality improvement with combined occipital and temporal signals, while adding parietal regions further enhances structural fidelity. These findings align with established visual processing pathways, supporting our dual-pathway architecture design.

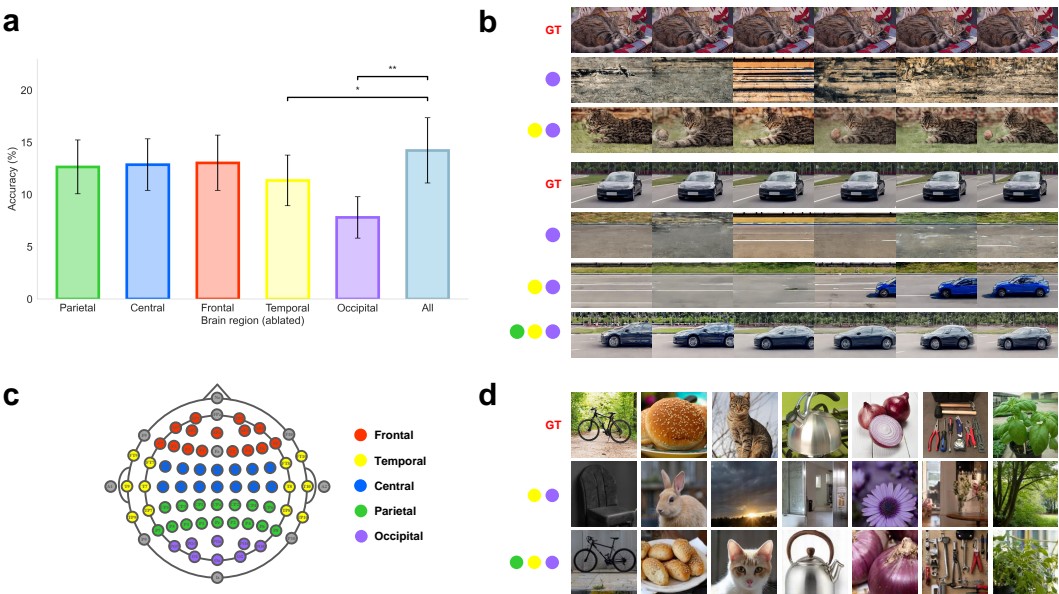

Figure 6: Brain region contribution analysis. (a) Classification accuracy with different brain regions ablated(SEED-DV). (b) Video reconstruction using different electrode combinations for static and dynamic scenes(SEED-DV). (c) EEG electrode distribution across five brain regions. (d) Static image reconstruction(THINGS-EEG).

## 5 Conclusion

In this paper, we introduce NEED, a unified framework addressing cross-subject variability and task-specificity in EEG-based visual reconstruction. Our approach maintains substantial performance when generalizing to unseen subjects and enables zero-shot transfer between video and image tasks. This work demonstrates how neural decoding can move beyond subject and task boundaries toward truly generalizable brain-computer interfaces. We provide further discussions on broader impacts and future work in Appendix G and H, respectively.

## 6 Limitations

Despite NEED's advances, certain challenges remain in EEG-based visual reconstruction. Reconstruction quality, while improved, still falls short of photorealistic fidelity for scenes with complex lighting and fine details. The framework currently performs optimally with higher-density EEG recordings, presenting challenges for deployment with ultra-portable consumer devices. These represent broader challenges in neural decoding rather than specific weaknesses of our approach.

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

# A  Related Work

## A.1  EEG-based Text Decoding

Brain-to-text decoding has evolved significantly as researchers explore non-invasive EEG-based methods. Willett et al. [59] established landmark performance in invasive systems using handwriting-based paradigms, while parallel non-invasive EEG approaches have been pursued for their practical advantages [7]. Murphy et al. [15] pioneered part-of-speech decoding from EEG, demonstrating that grammatical information could be extracted from non-invasive recordings.

Affolter et al. introduced Brain2Word [16], a foundational framework for decoding brain activity into natural language, while Wang and Ji [14] developed open vocabulary EEG-to-text decoders with zero-shot sentiment classification capabilities. Zou et al. [60] proposed cross-modal cloze tasks for brain-to-word decoding and later integrated pre-trained encoder-decoder models to improve performance [61].

Tang et al. [62] demonstrated semantic reconstruction of continuous language from non-invasive recordings, while Duan et al. [17] developed DEWave, a discrete encoding approach leveraging EEG wave properties for text translation. Makin et al. [63] established machine translation frameworks for cortical activity, and Défossez et al. [64] presented methods for decoding speech perception from non-invasive brain recordings. Recent advances include Jeong et al.'s [7] real-time deep neurolinguistic learning system and Moses et al.'s [32] neuroprosthesis for decoding speech in paralyzed individuals. These developments leverage innovations in EEG signal processing [65, 12, 11] and self-supervised approach [66] that have proven effective across neural decoding domains.

## A.2  EEG-based Image Decoding

EEG-based image reconstruction has advanced significantly through innovations in neural architectures and generative modeling. Early work by Palazzo et al. [1] established multimodal learning for decoding brain representations by integrating neural activity and visual features, providing a foundation for subsequent research. Song et al. [48] developed EEG Conformer, combining convolutional and transformer architectures to enhance spatial-temporal modeling of EEG signals, while Pan et al. [11] introduced MAtt, a manifold attention network designed to address the high-dimensional, non-linear nature of neural data.

Recent approaches have leveraged sophisticated generative models for improved reconstruction quality. Takagi and Nishimoto [67] demonstrated high-resolution image reconstruction using latent diffusion models from brain activity, while Lin et al. [22] presented Mind Reader for reconstructing complex images from EEG signals. These advances have been complemented by cross-subject generalization techniques, with Ho et al. [68] introducing hierarchical neural code conversion for inter-individual image reconstruction and Fang et al. [24] developing methods to alleviate the semantic gap in generalized brain-to-image reconstruction. Collectively, these innovations in architectures, generative modeling, and transfer learning have significantly enhanced the capabilities of EEG-based image reconstruction systems [21, 5, 23, 18, 44].

## A.3  EEG-based Emotion Recognition

EEG-based emotion recognition has evolved significantly with advancements in cross-subject generalization techniques. Zhou et al. [69] developed an Emotional EEG Style Transfer Network that enhances cross-dataset emotion recognition by normalizing subject-specific patterns, while Shen et al. [52] introduced contrastive learning of subject-invariant EEG representations for improved cross-subject performance. Li et al. [53] proposed dynamic domain adaptation for both cross-subject and cross-session emotion recognition, addressing the inherent variability in neural responses across individuals. Zhao et al. [51] created a plug-and-play domain adaptation framework that enables efficient adaptation to new subjects without extensive retraining, while Chen et al. [37] introduced Personal-zscore to normalize individual differences in signal characteristics.

These approaches have been evaluated across diverse applications, with Peng et al. [36] implementing joint EEG feature transfer with semisupervised learning, and Liu et al. [70] developing DA-CapsNet, a multi-branch capsule network leveraging adversarial domain adaptation. She et al. [35] focused on multisource associate domain adaptation for both cross-subject and cross-session scenarios, while Xu

et al. [71] addressed the challenge of heterogeneity in temporal scale among different cortical regions. Collectively, these innovations represent significant progress toward robust emotion recognition systems that can generalize effectively across individuals, sessions, and experimental conditions [33, 34, 41, 72, 70].

### A.4 EEG-based Cross-Task Learning

Cross-task learning has emerged as a critical approach for maximizing knowledge transfer in EEG-based applications. Zhou et al. [73] pioneered cognitive workload recognition across different tasks using domain adaptation techniques, while Guan et al. [74] developed tensor representation methods with transfer learning to improve cross-task mental workload detection. Mota et al. [75] introduced a deep descriptor specifically designed for cross-tasking EEG-based recognition that enables knowledge sharing between different cognitive paradigms. The effectiveness of these approaches has been demonstrated in various applications, with Pandey et al. [42] enhancing classification through domain adaptation with EEGNet architecture, and Liu et al. [43] establishing a generalized depression recognition framework based on cross-center and cross-task EEG signals.

### A.5 Cross-Subject Neural Decoding

Cross-subject generalization represents a fundamental challenge in neural decoding, with significant advances in both EEG and fMRI modalities. For EEG-based applications, Chen et al. [37] introduced Personal-zscore to eliminate individual differences through effective normalization of subject-specific signal characteristics, while Zhao et al. [51] developed a plug-and-play domain adaptation framework that enables efficient transfer to new subjects. Li et al. [53] proposed dynamic domain adaptation for both cross-subject and cross-session scenarios, addressing the critical challenge of temporal variability across recording sessions. These approaches have substantially improved generalization performance in various domains, including emotion recognition [52, 35, 36] and video reconstruction [27, 28].

In parallel, fMRI-based cross-subject decoding has seen remarkable progress through specialized alignment techniques. Wang et al. [76] developed MindBridge, a comprehensive framework that establishes common neural representations across individuals, while Li et al. [50] introduced global-local functional alignment for enhanced cross-subject fMRI-to-video decoding. Liu et al. [77] focused on learning transferable brain decoding models from cross-subject fMRI data, and Gong et al. [38] proposed MindTuner, which leverages visual fingerprinting and semantic correction for improved generalization. These innovations, supported by hierarchical modeling of functional brain networks [39], collectively demonstrate significant advancement toward overcoming the substantial individual variability that has traditionally limited the generalizability of neural decoding systems.

### A.6 Video-based Neural Decoding

Recent advances in video decoding from neural signals have demonstrated remarkable capabilities in reconstructing dynamic visual experiences. Liu et al. [26] developed EEG2video, a pioneering framework for decoding dynamic visual perception from EEG signals that effectively captures temporal coherence in video reconstruction. Yao et al. [27] identified temporal correlations between natural single-shot videos and EEG signals, while Song et al. [28] demonstrated asynchronous video target detection based on single-trial EEG recordings. These EEG-based approaches are complemented by Jang et al.'s [29] work on emotional video classification through learning connectivity structures, establishing the relationship between neural activity patterns and affective video content.

In parallel, fMRI-based video decoding has achieved high-fidelity reconstruction through specialized architectures. Chen et al. [30] introduced Cinematic Mindscapes for high-quality video reconstruction from brain activity, while Wang et al. [30] developed an fMRI-conditional video generative adversarial network for reconstructing rapid natural vision. Li et al. [50] enhanced cross-subject fMRI-to-video decoding through global-local functional alignment, addressing the critical challenge of individual variability. These advances have been supported by comprehensive datasets like BOLD Moments [49], which provides rich metadata for modeling short visual events, enabling researchers to establish increasingly sophisticated mappings between neural activity and dynamic visual experiences across both EEG and fMRI modalities.

# B    Algorithm

---

**Algorithm 1** Training and Inference of NEED Framework

---

**Require:**
 1: **Training:** EEG dataset $\mathcal{D}_{train} = \{x_i, v_i, d_i\}$, auxiliary datasets $\mathcal{D}_{aux}$, pretrained models $T2I, B, C$
 2: **Inference:** Unseen subject EEG $x_{test}$, task type $\tau_t \in \{\text{video}, \text{image}\}$
**Ensure:** Trained model components: $IAM, DSGNet, PU, SU$, Inference Mechanism
 3: **Training Stage:**
 4: Pretrain $IAM$ on multiple datasets: $\mathcal{L}_{pretrain} = \alpha\mathcal{L}_{recon} + \beta\mathcal{L}_{adv} + \gamma D_{KL}$
 5: **for** each $(x_i, v_i) \in \mathcal{D}_{train}$ **do**
 6:      Process EEG: $Z_i = IAM(x_i), Z_{DSG} = DSGNet(Z_i)$
 7: **end for**
 8: Train $PU$ with Bidirectional Dynamic Contrastive loss:
 9: $\mathcal{L}_{BDC} = -\sum_{i,j} w_{ij} \log \frac{\exp(sim(Z_0^i, V_1^j)/\tau)}{\sum_k \exp(sim(Z_0^i, V_1^k)/\tau)} + \lambda\|\nabla_t Z_0 - \nabla_t V_1\|_2^2$
10: Train $SU$ with Hierarchical Semantic Contrastive loss:
11: $\mathcal{L}_{HSC} = \sum_{l=1}^{L} \alpha_l \cdot \left[-\log \frac{\exp(Z_s^l \cdot C_k^l/\tau)}{\sum_n \exp(Z_s^l \cdot C_n^l/\tau)}\right] + \beta \cdot \|G(Z_s) - G(C_k)\|_F$
12: Train unified inference with Cross-modal Alignment and Diffusion:
13: $\mathcal{L}_{CSA} = \mathcal{L}_{cont}(Z_3, T) + \gamma \cdot \mathcal{L}_{KL}(Z_3, T) + \delta \cdot \mathcal{L}_{align}(Z_3, \text{Keyframe})$
14: Learn task-specific parameters $\gamma_l(\tau_t)$ and $\beta_l(\tau_t)$ for conditional layer modulation
15: $h_l(x_t, t, c_t, \tau_t) = \gamma_l(\tau_t) \odot \text{Norm}(h_{l-1}) + \beta_l(\tau_t)$
16: **Inference Stage:**
17: Process test EEG: $Z_{test} = IAM(x_{test}), Z_{DSG} = DSGNet(Z_{test})$
18: Extract perception features: $Z_p = PU(Z_{DSG})$
19: Extract semantic features: $Z_s = SU(Z_{DSG})$
20: Apply task adaptation: $Z_{task} = \text{TaskAdapter}([Z_p; Z_s], \tau_t)$
21: Configure conditional weights based on task: $\alpha_{temp} = \sigma(W_\tau \cdot \tau_t), \alpha_{spat} = \sigma(W_\tau \cdot \tau_t)$
22: Generate integrated conditioning:
23: $c_t = \alpha_t \cdot \text{ControlNet}(\text{Keyframe}_{pred}) + \beta_t \cdot \text{TextEmbed}(\text{Caption}) + \gamma_t \cdot \text{DynPattern} + \delta_t \cdot Z_{eeg}$
24: Initialize: $x_T \sim \mathcal{N}(0, I)$
25: **for** $t = T$ to $1$ **do**
26:      Apply task-adaptive denoising: $x_{t-1} = \mu_\theta(x_t, t, c_t, \tau_t) + \sigma_t \cdot z$
27:      Apply conditional layer modulation: $h_l(x_t, t, c_t, \tau_t) = \gamma_l(\tau_t) \odot \text{Norm}(h_{l-1}) + \beta_l(\tau_t)$
28: **end for**
29: **Return:** Reconstructed visual content $x_0$ (video or image based on $\tau_t$)

---

# C    Extended Method Details

## C.1    Hierarchical Individual Adaptation Module

The Individual Adaptation Module (IAM) is central to our framework's cross-subject generalization capability. Here we provide a detailed description of its multi-level adaptation architecture and working principles.

### C.1.1    Multi-Level Adaptation Architecture

As shown in Fig 7, IAM implements a hierarchical adaptation approach that processes EEG signals at four progressively abstract levels: signal level, spectral level, temporal level, and semantic level. This design allows the module to address different aspects of inter-subject variability simultaneously.

**Signal-Level Adaptation.** At the lowest level, IAM normalizes physical characteristics of EEG signals that vary across subjects due to differences in skull thickness, electrode placement, and skin conductivity. For each subject $s$, we learn transformation parameters:

$$E_{signal} = W_{signal}(s) \cdot E_{original} + b_{signal}(s) \tag{6}$$

where $W_{signal}(s) \in \mathbb{R}^{C \times C}$ is a subject-specific transformation matrix that recalibrates channel relationships, and $b_{signal}(s) \in \mathbb{R}^C$ is a bias term that adjusts baseline activity. For subjects with

different numbers of channels than our reference configuration, the transformation includes an additional projection step that maps between different dimensionalities.

**Spectral-Level Adaptation.** Individual differences in oscillatory patterns are well-documented in EEG literature, with substantial variability in the power and frequency characteristics of canonical brain rhythms. Our spectral adaptation addresses this by learning subject-specific weights for different frequency bands:

$$E_{spectral} = \sum_{band \in \{\delta, \theta, \alpha, \beta, \gamma\}} w_{band}(s) \cdot \text{BPF}_{band}(E_{signal}) \tag{7}$$

where $\text{BPF}_{band}$ represents bandpass filtering for canonical frequency bands, and $w_{band}(s)$ are subject-specific importance weights. The filtered signals are further processed through a subject-conditioned spectral attention mechanism:

$$A_{spectral}(f) = \text{softmax}(v_s^T \tanh(W_s \cdot \text{PSD}(E_{spectral}) + b_s)) \tag{8}$$

where PSD computes the power spectral density, and $v_s$, $W_s$, and $b_s$ are subject-specific parameters.

**Temporal-Level Adaptation.** Different subjects exhibit varying temporal dynamics in their neural responses, including latency differences and distinct temporal patterns. To address this, IAM employs a subject-conditioned recurrent module:

$$E_{temporal} = h_t = \text{GRU}(E_{spectral}, h_{t-1}; \theta_{temporal}(s)) \tag{9}$$

where $\theta_{temporal}(s)$ represents subject-specific GRU parameters. To reduce parameter overhead, we implement this through hypernetworks where a small subject embedding generates the modification to base GRU parameters:

$$\theta_{temporal}(s) = \theta_{base} + \text{MLP}(s_{embed}) \tag{10}$$

**Semantic-Level Adaptation.** At the highest level, IAM processes cognitive patterns that vary across subjects through our subject-aware attention mechanism:

$$Z_{attn} = \sum_{h=1}^{H} W_O^h \left( \sigma \left( \frac{(E_i W_Q^h + s_i U_Q^h)(E_i W_K^h + s_i U_K^h)^T}{\sqrt{d_k}} \right) (E_i W_V^h) \right) \tag{11}$$

This mechanism allows the model to attend differently to EEG features based on subject-specific characteristics, capturing individual differences in neural encoding of visual information.

While the multi-level adaptation captures subject-specific patterns, the subject-invariant transformation $\Gamma_\phi$ extracts common neural patterns across subjects:

$$\Gamma_\phi(E_i, \Omega_S, d_j) = \text{MHA} \left( \text{LN} \left( \text{FiLM} \left( E_i, W_\phi^T d_j + b_\phi^T d_j \right), \Omega_S \right) \right) \tag{12}$$

where MHA denotes multi-head attention, LN is layer normalization, and FiLM represents feature-wise linear modulation with learned weight matrix $W_\phi$ and bias vector $b_\phi$. The dataset embedding $d_j$ allows the transformation to adapt to different experimental paradigms, while $\Omega_S$ represents subject-agnostic knowledge learned during pretraining.

### C.1.2 Adaptive Integration Mechanism

The final normalized representation integrates information from all adaptation levels through a dynamic gating mechanism:

$$[g_{signal}, g_{spectral}, g_{temporal}, g_{attn}, g_\Gamma] = \text{softmax}(\text{MLP}([E_{signal}; E_{spectral}; E_{temporal}; Z_{attn}; \Gamma_\phi])) \tag{13}$$

$$Z_{final} = g_{signal} \cdot E_{signal} + g_{spectral} \cdot E_{spectral} + g_{temporal} \cdot E_{temporal} + g_{attn} \cdot Z_{attn} + g_\Gamma \cdot \Gamma_\phi \tag{14}$$

This adaptive weighting allows IAM to emphasize different adaptation levels based on input characteristics, providing flexibility across diverse EEG patterns.

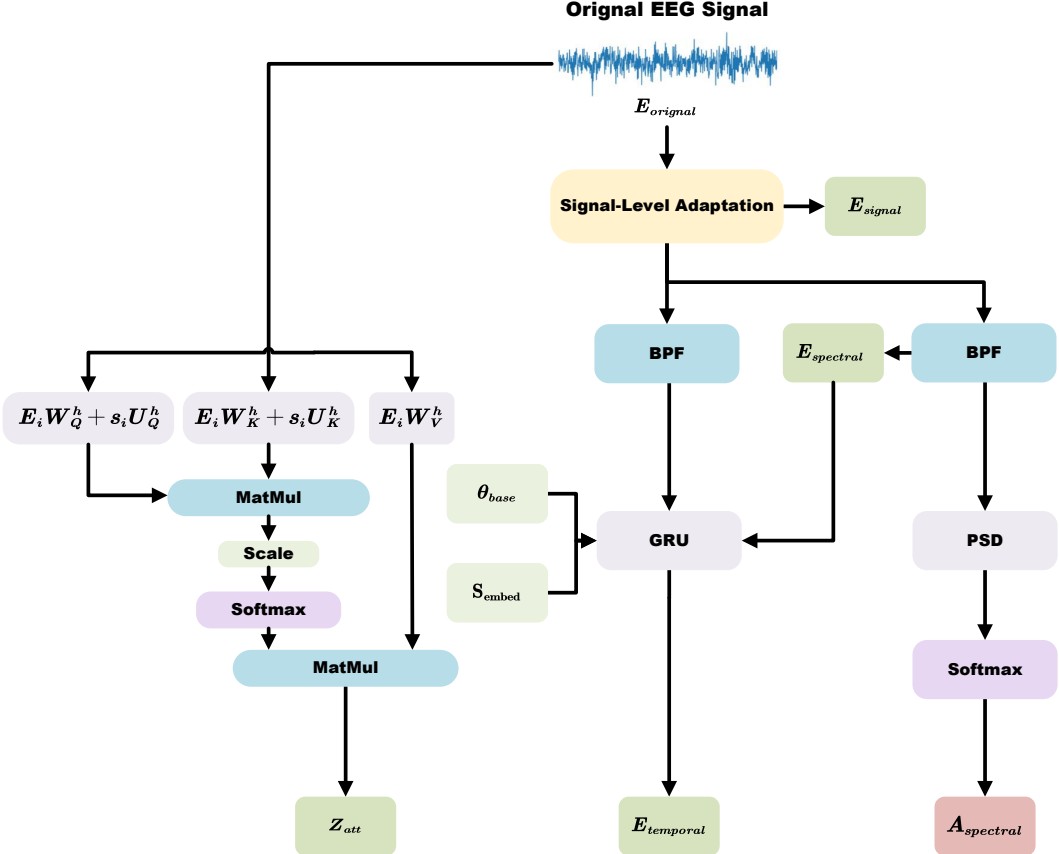

Figure 7: Multi-Level Adaptation Architecture

### C.1.3 Adversarial Training Strategy

To further enhance subject invariance in the normalized representations, we employ an adversarial training strategy. During pretraining, a subject discriminator $D$ attempts to identify subject identity from the normalized representations, while IAM is trained to generate representations that maximize reconstruction quality while minimizing the discriminator's performance:

$$\mathcal{L}_{adv}(\theta_D, \theta_G) = \mathbb{E}_{Z_{norm} \sim p_{Z|s}}[\log D(Z_{norm})] + \mathbb{E}_{Z_{norm} \sim p_Z}[\log(1 - D(Z_{norm}))] \quad (15)$$

This adversarial objective is balanced with reconstruction quality and KL divergence terms:

$$\mathcal{L}_{pretrain} = \alpha \mathcal{L}_{recon} + \beta \mathcal{L}_{adv}(\theta_D, \theta_G) + \gamma D_{KL}(p(Z_{norm}|S)\|p(Z_{norm})) \quad (16)$$

### C.1.4 Implementation Details

The IAM architecture uses 6 attention heads with embedding dimension 256 for the subject-aware attention mechanism. Subject embeddings are initialized randomly and learned during training, with dimension $d_s = 64$. For the spectral processing, we use standard frequency bands: delta (0.5-4 Hz), theta (4-8 Hz), alpha (8-13 Hz), beta (13-30 Hz), and gamma (30-100 Hz).

The module is implemented with layer normalization and residual connections throughout to stabilize training. Dropout (rate 0.2) is applied after each major component to prevent overfitting to subject-specific patterns. The adaptive integration mechanism uses a 2-layer MLP with hidden dimension 128 and LeakyReLU activation.

During pretraining, we use AdamW optimizer with learning rate 1e-4 and weight decay 0.01. The loss function weights are set to $\alpha = 1.0$, $\beta = 0.5$, and $\gamma = 0.1$ to balance reconstruction quality with subject disentanglement.

## C.2 Dual-Stream EEG Encoder Details

Here we provide detailed mathematical formulations of our Dual-Stream EEG Encoder (DSGNet), which forms the foundation of our framework.

### C.2.1 Spatial Stream Details

The spatial stream begins with spherical harmonic representation of electrode positions. For electrodes located at spherical coordinates $(\theta_i, \phi_i)$ on the scalp, the initial spatial embedding is:

$$E_{spatial}^{(0)}(i) = \sum_{l=0}^{L} \sum_{m=-l}^{l} c_{l,m} Y_l^m(\theta_i, \phi_i) \tag{17}$$

where $Y_l^m$ are spherical harmonics of degree $l$ and order $m$, and $c_{l,m}$ are learnable coefficients. The geodesic-based adjacency matrix $A_{ij} = \exp(-d_{ij}^2/2\sigma^2)$ captures inter-electrode relationships, which are processed through graph convolutions:

$$E_{spatial}^{(l+1)} = \sigma \left( \tilde{D}^{-\frac{1}{2}} \tilde{A} \tilde{D}^{-\frac{1}{2}} E_{spatial}^{(l)} W^{(l)} \right) + \text{BatchNorm} \left( E_{spatial}^{(l)} \right) \tag{18}$$

where $\tilde{A} = A + \lambda I_N$ incorporates self-connections and $\tilde{D}_{ii} = \sum_j \tilde{A}_{ij}$ is the degree matrix.

### C.2.2 Temporal Stream Details

The temporal stream implements a discrete approximation of the Riemann-Liouville fractional transformation:

$$\mathcal{R}^\alpha(\mathbf{X})_t \approx \frac{\Delta t^\alpha}{\Gamma(\alpha+1)} \sum_{k=0}^{\lfloor t/\Delta t \rfloor} \omega_k^{(\alpha)} \mathbf{X}(t - k\Delta t) \tag{19}$$

where $\omega_k^{(\alpha)} = \frac{\Gamma(k+\alpha)}{\Gamma(k+1)\Gamma(\alpha)}$ are fractional binomial coefficients. This multi-scale approach with $\alpha \in \{0.2, 0.4, 0.6, 0.8\}$ captures temporal dependencies at various scales. Dilated temporal convolutions further expand the receptive field:

$$\mathcal{F}^{(l)}(t) = \sum_{k=0}^{K-1} f^{(l)}(k) \cdot \mathcal{F}^{(l-1)}(t - d^{(l)} \cdot k) \tag{20}$$

with kernel size $K = 5$ and dilation factor $d^{(l)} = 2^{l-1}$ for layer $l$, yielding an effective receptive field of 31 time steps with 4 layers.

### C.2.3 Cross-Stream Integration Details

The cross-stream attention implements multi-head attention with $h = 8$ heads:

$$Z_{cross} = \text{Concat} \left( \text{head}_1, \ldots, \text{head}_h \right) W^O \tag{21}$$

where $\text{head}_i = \text{Attention}(E_{spatial} W_i^Q, E_{temporal} W_i^K, E_{temporal} W_i^V)$ and $\text{Attention}(Q, K, V) = \text{softmax}(QK^T/\sqrt{d_k})V$.

The gated fusion mechanism combines features through:

$$Z = \sigma(W_g \cdot [E_{spatial}; E_{temporal}; Z_{cross}] + b_g) \odot \tanh(W_f \cdot [E_{spatial}; E_{temporal}; Z_{cross}] + b_f) \tag{22}$$

Our multi-dimensional masking strategy applies three types of masks: $M_{spatial}$ (simulating electrode disconnections), $M_{temporal}$ (temporal signal loss), and $M_{frequency}$ (frequency-band noise), sampled

from Bernoulli distributions with parameters $p_{spatial} = 0.8$, $p_{temporal} = 0.9$, and $p_{frequency} = 0.85$ respectively. The final representation is computed as $\tilde{E} = Z \odot M_{spatial} \odot M_{temporal} \odot M_{frequency}$.

## C.3 Cross-Dataset Channel Adaptation

A fundamental challenge in EEG-based neural decoding is addressing the heterogeneity of acquisition systems. Our NEED framework implements a comprehensive channel adaptation strategy that enables seamless generalization across varied EEG configurations. This approach is critical for practical applications where data might come from different hardware setups.

Our unified coordinate system aligns all electrode locations to a standardized spherical model based on the international 10-10 system. For similar channel setups, we establish direct correspondence between electrodes based on spatial proximity. Fig 8 illustrates our mapping approach across different datasets.

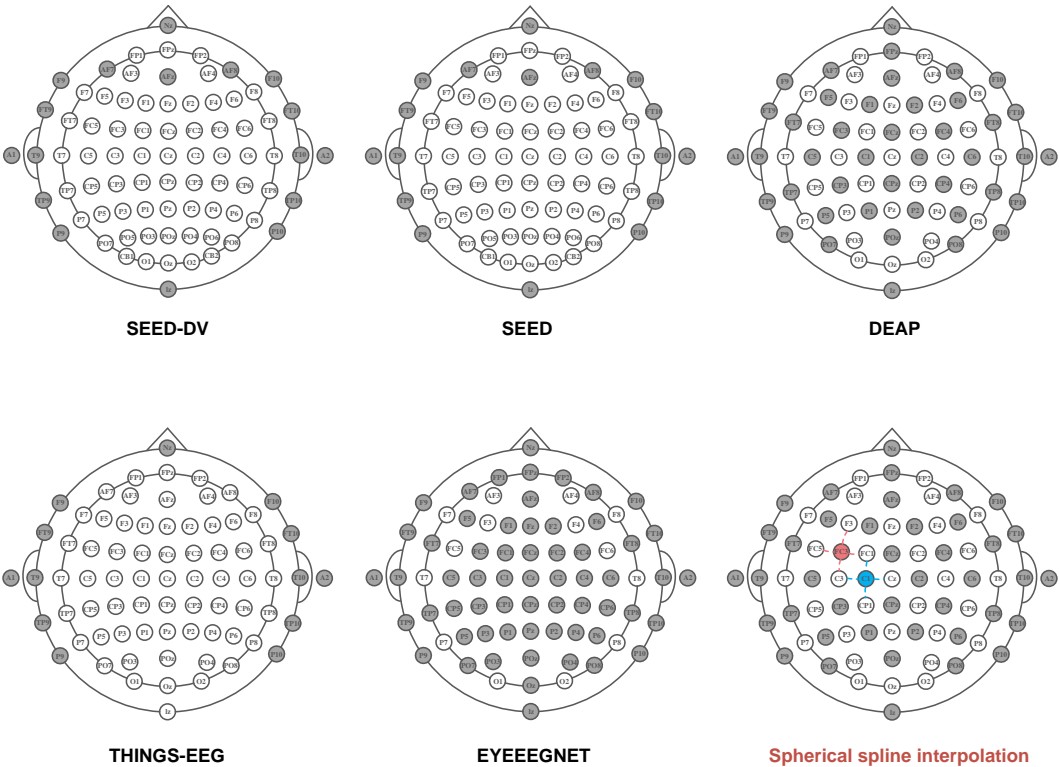

Figure 8: Electrode mapping visualization across different EEG datasets. The figure shows the spatial distribution of electrodes in SEED-DV, SEED, DEAP, THINGS-EEG, and EEGEYENET datasets. The rightmost visualization demonstrates spherical spline interpolation used to normalize channel configurations.

When transferring to systems with fewer channels (e.g., from 62 to 14 channels), we prioritize electrodes covering key visual processing regions. Conversely, for adaptation to higher-density configurations, we employ spherical spline interpolation to reconstruct signals at missing locations:

$$\hat{E}_j = \sum_{i \in C} w_{ij} E_i \tag{23}$$

where $\hat{E}_j$ is the interpolated value at channel location $j$ in the reference configuration, $E_i$ is the measured value at existing channel $i$, and $w_{ij}$ are weights based on the spherical distance between locations:

$$w_{ij} = \frac{\exp(-d_{ij}^2/2\sigma^2)}{\sum_{k \in C} \exp(-d_{kj}^2/2\sigma^2)} \tag{24}$$

The parameter $\sigma$ controls the influence radius of each electrode and is set to 0.15 radians (approximately 8.6 degrees on the scalp surface) based on empirical validation across datasets.

Beyond spatial mapping, we address spectral differences through unified preprocessing: resampling all data to 200Hz, applying a consistent 0.5-40Hz bandpass filter, and implementing frequency band-specific gain adjustments. This approach normalizes signal characteristics across acquisition systems:

$$E_{normalized}(f) = G(f) \cdot E_{original}(f) \tag{25}$$

where $G(f)$ is a frequency-dependent gain function derived from comparing power spectral density distributions across datasets. The gain functions are defined per frequency band to account for hardware-specific filtering effects:

$$G(f) = \begin{cases} G_\delta, & \text{if } f \in [0.5, 4 \text{ Hz}] \\ G_\theta, & \text{if } f \in [4, 8 \text{ Hz}] \\ G_\alpha, & \text{if } f \in [8, 13 \text{ Hz}] \\ G_\beta, & \text{if } f \in [13, 30 \text{ Hz}] \\ G_\gamma, & \text{if } f \in [30, 40 \text{ Hz}] \end{cases} \tag{26}$$

The IAM module integrates channel importance learning, automatically adapting to available electrodes by assigning importance weights ($\alpha_i$) during training:

$$Z_{channel} = \sum_{i=1}^{C} \alpha_i \cdot E_i \tag{27}$$

These weights are initialized based on neurophysiological priors (higher weights for occipital and temporal regions for visual tasks) but evolve during training to optimize cross-dataset generalization. The importance weights are learned through an attention mechanism:

$$\alpha_i = \frac{\exp(v^T \tanh(W_c E_i + b_c))}{\sum_{j=1}^{C} \exp(v^T \tanh(W_c E_j + b_c))} \tag{28}$$

where $v$, $W_c$, and $b_c$ are learned parameters. This approach enables our framework to process EEG from diverse recording systems within a unified computational architecture, enabling true cross-dataset compatibility.

## D    Datasets and Preprocessing

Our NEED framework leverages multiple EEG datasets in a strategic manner to enable zero-shot cross-subject and cross-task generalization. Table 3 provides an overview of these datasets and their roles in our framework.

Table 3: Overview of EEG datasets and their roles in the NEED frame work.

| Dataset | Subjects | EEG Channels | Stimuli Type | Role in NEED |
|---|---|---|---|---|
| SEED-DV | 20 | 62 | Video clips (40 classes) | Primary training for video reconstruction |
| SEED | 15 | 62 | Emotional film clips | IAM pretraining (dynamic stimuli) |
| DEAP | 32 | 32 | Music videos | IAM pretraining (affective responses) |
| THINGS-EEG | 10 | 64 | Static images (1,854 concepts) | Evaluation for cross-task generalization |
| EEGEYENET | 356 | 14 | Various visual tasks | Evaluation for cross-subject generalization |

## D.1 SEED-DV Dataset

SEED-DV (SJTU Emotion EEG Dataset - Dynamic Video) serves as our primary training dataset for video reconstruction. It contains high-density EEG recordings from 20 subjects watching 1,400 video clips across 40 visual categories, as shown in table 4. Each 2-second clip is presented at 24 frames per second with a resolution of $1920 \times 1080$ pixels. EEG data is recorded using a 62-channel cap according to the international 10-10 system, sampled at 1000 Hz using ESI NeuroScan System.

Table 4: Visual Categories in the SEED-DV Dataset

| SEED-DV Dataset: 40 Visual Categories | | | | |
|---|---|---|---|---|
| **Food Items** | **Animals** | **Human Activities** | **Natural Scenes** | **Dynamic Events** |
| Watermelon | Turtle | Skiing | Mountain | Fireworks |
| Pizza | Shark-like | Running | Buildings | Water |
| Drink | Jellyfish | Dancing | Beach | Car |
| Cake | Fish/Coral | Boxing | Tree | Hot balloon |
| Banana | Bird | Face | Mushroom | Airplane |
| | Rabbit | Crowd | Flower | Ship |
| | Panda | Couple | Road | Bike |
| | Horse | | | Piano |
| | Elephant | | | Guitar |
| | Dog | | | Drum |
| | Cat | | | |

The visual stimuli in SEED-DV include a diverse range of content categories: natural scenes (e.g., mountains, oceans, forests), urban environments (e.g., city streets, buildings, traffic), human activities (e.g., sports, dancing, cooking), animal behaviors, and various object interactions. This diversity is crucial for training a robust visual decoder capable of reconstructing a wide spectrum of visual experiences.

For preprocessing, we apply a bandpass filter (0.5-40 Hz) to remove artifacts, downsample to 200 Hz, and segment the continuous EEG into 2-second epochs corresponding to individual video clips. Independent Component Analysis (ICA) is used to remove eye movement and muscle artifacts. The cleaned EEG signals are then normalized per channel using z-score normalization.

## D.2 SEED Dataset

The SEED dataset contains EEG recordings from 15 subjects watching emotional film clips designed to elicit specific emotional states (positive, neutral, and negative). EEG is recorded using the same 62-channel system as SEED-DV. While originally developed for emotion recognition, we leverage this dataset during IAM pretraining to capture neural responses to dynamic visual stimuli with varying emotional content.

The film clips in SEED range from 4-6 minutes in length and include excerpts from popular movies and documentaries that have been validated to reliably induce the target emotions. This dataset helps our model generalize across different types of dynamic visual content with varied emotional significance.

## D.3 DEAP Dataset

DEAP (Database for Emotion Analysis using Physiological Signals) contains EEG recordings from 32 subjects watching 40 one-minute music videos. EEG is recorded using 32 channels, providing complementary information about neural responses to audiovisual stimuli. We use this dataset during IAM pretraining to enhance cross-subject generalization capabilities.

Although DEAP uses fewer EEG channels (32) compared to our primary datasets, its larger subject pool provides valuable information about inter-subject variability. The IAM module learns to normalize these diverse patterns into a common representation space through our multi-dataset knowledge transfer approach.

### D.4 THINGS-EEG Dataset

The THINGS-EEG dataset contains EEG recordings from 10 subjects viewing 22,000 static images across 1,854 object categories. EEG is recorded using a 64-channel system. Importantly, this dataset uses static image stimuli rather than videos, making it ideal for evaluating cross-task generalization.

The THINGS image collection contains diverse everyday objects photographed under various conditions. Each stimulus is presented for 500ms followed by a 1000ms inter-stimulus interval. We extract epochs corresponding to the stimulus presentation for our cross-task evaluation. The high diversity of object categories in this dataset provides a rigorous test for semantic reconstruction capabilities.

### D.5 EEGEYENET Dataset

EEGEYENET is the largest publicly available EEG dataset for visual tasks, containing recordings from 356 subjects performing various visual tasks including object recognition, scene categorization, and visual search. EEG is recorded using a portable 14-channel system (EMOTIV EPOC+), making it substantially different from our training datasets in terms of channel configuration and signal quality.

This dataset provides an extreme test of our framework's ability to generalize across subjects, particularly with significantly different EEG acquisition systems. The 14-channel configuration poses a particular challenge for spatial pattern extraction, requiring our IAM to effectively normalize these sparse recordings into the common representation space learned from denser EEG montages.

### D.6 Data Preprocessing

We preprocess EEG data using standard filtering (0.5-40 Hz), downsampling to 200 Hz, and segmenting the continuous EEG into 2-second epochs corresponding to individual visual stimuli. Independent Component Analysis (ICA) is applied to remove eye movement and muscle artifacts. The cleaned EEG signals are then normalized per channel using z-score normalization. For SEED-DV specifically, we apply additional preprocessing steps to account for the high temporal resolution of video stimuli, including temporal alignment of EEG with visual frame transitions and removal of inter-trial noise. Details of our cross-dataset channel adaptation approach, which addresses the varying number of EEG channels across datasets, are provided in Appendix C.3.

## E  Implementation Details

We implement our framework using PyTorch and conduct experiments on NVIDIA A100 GPUs. The Individual Adaptation Module (IAM) employs a multi-head attention mechanism (6 heads, dimension 256) with subject embeddings (dimension 64) and dataset-conditional modulation. Our Dual-Stream EEG Encoder (DSGNet) integrates spatial processing using spherical harmonic representations and 4-layer graph neural networks, with temporal processing through multi-scale fractional transformations at alpha parameters {0.2, 0.4, 0.6, 0.8}.

The Perception Understanding (PU) module implements a three-branch architecture extracting complementary visual features: motion patterns through 3D networks, spatial structures via 2D networks, and temporal dynamics using our SMARN architecture (2 layers, hidden size 256). The Semantic Understanding (SU) module adapts CLIP-ViT-G/14 and LLaMA-2-13B with LoRA fine-tuning (rank 16) for EEG feature translation. Training employs the AdamW optimizer (learning rates 1e-4 to 2e-5) with weight decay (0.01) and cosine scheduling.

For video reconstruction, we extract 6 keyframes per 2-second clip and employ temporal attention mechanisms to ensure coherence. For image reconstruction, we adapt the same framework but modify the conditioning signals with different weighting coefficients ([0.3, 0.4, 0.1, 0.2] versus [0.4, 0.3, 0.2, 0.1] for videos). The unified inference mechanism uses classifier-free guidance (scale 7.5) with 50 denoising steps.

We follow different evaluation protocols for different generalization scenarios: 5-fold cross-validation for within-subject assessment, leave-one-out testing for cross-subject generalization, and zero-shot transfer (without fine-tuning) for cross-task evaluation. This enables comprehensive assessment of our framework's generalization capabilities across different dimensions.

# F Supplementary Results

## F.1 Ablation results

### F.1.1 PU Ablation results

Table 5: Detailed ablation study on Perception Understanding (PU) module components. This analysis examines how different components of the PU module affect temporal coherence and visual fidelity in reconstruction tasks.

| PU Module Configuration | Video Reconstruction (SEED-DV) | | | | Image Reconstruction (THINGS-EEG) | | | |
|---|---|---|---|---|---|---|---|---|
| | Temporal Metrics | | Visual Quality | | Structural Metrics | | Visual Detail | |
| | CLIP-pcc↑ | FVD↓ | 2-way Acc.↑ | SSIM↑ | Edge-F1↑ | Struct-sim↑ | LPIPS↓ | PixCorr↑ |
| **Full PU Module** | **0.738±0.04** | **175.3±8.6** | **0.839±0.03** | **0.356±0.02** | **0.427±0.03** | **0.493±0.02** | **0.387±0.03** | **0.162±0.01** |
| *1. Core Component Analysis* | | | | | | | | |
| *w/o Dynamic Feature Extractor (DFE)* | 0.485±0.05 | 283.7±15.8 | 0.713±0.04 | 0.231±0.03 | 0.315±0.04 | 0.356±0.03 | 0.528±0.04 | 0.103±0.02 |
| *w/o Spatiotemporal Memory-Augmented RNN* | 0.542±0.04 | 254.8±12.7 | 0.752±0.04 | 0.267±0.03 | 0.359±0.03 | 0.402±0.03 | 0.463±0.04 | 0.125±0.02 |
| *w/o Bidirectional Dynamic Contrastive (BDC)* | 0.617±0.05 | 218.6±11.3 | 0.787±0.03 | 0.298±0.03 | 0.385±0.04 | 0.438±0.03 | 0.422±0.03 | 0.143±0.02 |
| *2. DFE Component Analysis* | | | | | | | | |
| *w/o Motion Branch (Fm)* | 0.563±0.05 | 247.2±13.5 | 0.776±0.04 | 0.274±0.03 | 0.362±0.04 | 0.413±0.03 | 0.459±0.04 | 0.129±0.02 |
| *w/o Spatial Branch (Fs)* | 0.685±0.04 | 202.6±10.8 | 0.795±0.03 | 0.312±0.03 | 0.372±0.03 | 0.426±0.03 | 0.429±0.03 | 0.138±0.02 |
| *w/o Temporal Branch (Ft)* | 0.595±0.05 | 234.8±12.4 | 0.783±0.04 | 0.289±0.03 | 0.368±0.04 | 0.419±0.03 | 0.443±0.04 | 0.135±0.02 |
| *Motion + Spatial only (no Temporal)* | 0.628±0.05 | 222.3±11.5 | 0.792±0.03 | 0.301±0.03 | 0.376±0.03 | 0.432±0.03 | 0.428±0.03 | 0.141±0.02 |
| *Motion + Temporal only (no Spatial)* | 0.694±0.04 | 195.8±9.3 | 0.802±0.03 | 0.328±0.03 | 0.382±0.04 | 0.448±0.02 | 0.412±0.03 | 0.148±0.02 |
| *Spatial + Temporal only (no Motion)* | 0.652±0.05 | 214.7±10.6 | 0.798±0.04 | 0.315±0.03 | 0.379±0.03 | 0.441±0.03 | 0.418±0.03 | 0.145±0.02 |
| *3. SMARN Architecture Analysis* | | | | | | | | |
| *LSTM instead of SMARN* | 0.643±0.05 | 216.9±11.2 | 0.795±0.03 | 0.309±0.03 | 0.375±0.04 | 0.436±0.03 | 0.421±0.04 | 0.142±0.02 |
| *GRU instead of SMARN* | 0.632±0.04 | 219.5±10.8 | 0.791±0.04 | 0.305±0.03 | 0.372±0.03 | 0.430±0.03 | 0.425±0.03 | 0.140±0.02 |
| *Transformer instead of SMARN* | 0.681±0.05 | 203.8±9.7 | 0.813±0.03 | 0.327±0.02 | 0.392±0.03 | 0.457±0.03 | 0.405±0.03 | 0.153±0.01 |
| *w/o Hierarchical Memory* | 0.686±0.04 | 201.2±9.5 | 0.815±0.03 | 0.329±0.02 | 0.395±0.03 | 0.460±0.02 | 0.402±0.03 | 0.154±0.01 |
| *w/o Attentional Weights* | 0.703±0.04 | 193.4±9.1 | 0.821±0.03 | 0.337±0.02 | 0.403±0.03 | 0.469±0.02 | 0.398±0.03 | 0.156±0.01 |
| *4. Bidirectional Dynamic Contrastive (BDC) Analysis* | | | | | | | | |
| *w/o Temporal Gradient Term* | 0.689±0.04 | 195.3±9.4 | 0.817±0.03 | 0.333±0.02 | 0.399±0.03 | 0.464±0.02 | 0.401±0.03 | 0.155±0.01 |
| *w/o Learnable Attention Weights* | 0.702±0.04 | 192.7±9.2 | 0.823±0.03 | 0.339±0.02 | 0.405±0.03 | 0.471±0.02 | 0.396±0.03 | 0.157±0.01 |
| *Cosine vs. Euclidean Distance* | 0.728±0.04 | 183.4±8.8 | 0.831±0.03 | 0.347±0.02 | 0.417±0.03 | 0.486±0.02 | 0.392±0.03 | 0.159±0.01 |
| *Different Temperature (=0.05)* | 0.723±0.04 | 185.7±9.0 | 0.829±0.03 | 0.345±0.02 | 0.415±0.03 | 0.483±0.02 | 0.394±0.03 | 0.158±0.01 |
| *Different Temperature (=0.5)* | 0.717±0.05 | 188.2±9.1 | 0.825±0.03 | 0.341±0.02 | 0.411±0.03 | 0.478±0.03 | 0.397±0.03 | 0.156±0.01 |
| *5. Temporal Modeling Approaches* | | | | | | | | |
| *3D ConvNets instead of SMARN* | 0.698±0.04 | 194.2±9.3 | 0.819±0.03 | 0.334±0.02 | 0.401±0.03 | 0.467±0.02 | 0.400±0.03 | 0.156±0.01 |
| *Temporal Attention instead of SMARN* | 0.706±0.04 | 191.8±9.2 | 0.823±0.03 | 0.338±0.02 | 0.405±0.03 | 0.472±0.02 | 0.396±0.03 | 0.157±0.01 |
| *SlowFast Network instead of SMARN* | 0.712±0.04 | 189.5±9.0 | 0.826±0.03 | 0.342±0.02 | 0.409±0.03 | 0.476±0.02 | 0.394±0.03 | 0.158±0.01 |
| *TimeSformer instead of SMARN* | 0.721±0.04 | 186.3±8.9 | 0.830±0.03 | 0.347±0.02 | 0.415±0.03 | 0.484±0.02 | 0.391±0.03 | 0.160±0.01 |
| *6. Visual Feature Extractors* | | | | | | | | |
| *ResNet-50 instead of T3D/T2D/TTD* | 0.687±0.05 | 198.5±9.5 | 0.812±0.03 | 0.327±0.03 | 0.396±0.03 | 0.461±0.02 | 0.407±0.03 | 0.153±0.01 |
| *ViT-B/16 instead of T3D/T2D/TTD* | 0.702±0.04 | 192.6±9.2 | 0.823±0.03 | 0.339±0.02 | 0.405±0.03 | 0.472±0.02 | 0.396±0.03 | 0.157±0.01 |
| *Swin-T instead of T3D/T2D/TTD* | 0.714±0.04 | 188.9±9.0 | 0.828±0.03 | 0.343±0.02 | 0.410±0.03 | 0.477±0.02 | 0.393±0.03 | 0.158±0.01 |
| *ConvNeXt-T instead of T3D/T2D/TTD* | 0.719±0.04 | 186.7±8.9 | 0.829±0.03 | 0.345±0.02 | 0.413±0.03 | 0.481±0.02 | 0.392±0.03 | 0.159±0.01 |
| *7. Integration with Semantic Understanding* | | | | | | | | |
| *Late Fusion (PU+SU at end)* | 0.721±0.04 | 187.2±8.9 | 0.829±0.03 | 0.346±0.02 | 0.414±0.03 | 0.482±0.02 | 0.392±0.03 | 0.159±0.01 |
| *Feature-level Fusion* | 0.726±0.04 | 183.8±8.8 | 0.833±0.03 | 0.349±0.02 | 0.418±0.03 | 0.488±0.02 | 0.390±0.03 | 0.160±0.01 |
| *Hierarchical Fusion (ours)* | **0.738±0.04** | **175.3±8.6** | **0.839±0.03** | **0.356±0.02** | **0.427±0.03** | **0.493±0.02** | **0.387±0.03** | **0.162±0.01** |

As shown in Table 5, we conducted an importance evaluation of the submodules of the PU module on two different datasets: SEED-DV for video reconstruction and THINGS-EEG for image reconstruction. The table is divided into seven modules, each demonstrating the significance of different branches within the seven submodules of the PU module. The first section focuses on the three core components of the PU module: the Dynamic Feature Extractor (DFE), the Spatiotemporal Memory-Enhanced Recurrent Network (SMARN), and the Bidirectional Dynamic Contrastive Alignment Module (BDC).

When the DFE branch is removed, the Structural Similarity Index (SSIM) for the reconstruction task decreases by 0.125, which is twice the decrease observed when the BDC branch is removed. The quality of the model's reconstruction output significantly degrades, highlighting the crucial role of the DFE module in output similarity for reconstruction tasks. Further analysis reveals that the DFE module consists of three main branches: motion, spatial, and temporal. The motion branch has the most significant impact, with an SSIM drop of 0.082 after its removal, accounting for 66% of the overall decrease. This validates the core contribution of the motion branch in dynamic EEG decoding.

In terms of spatiotemporal sequence processing, we replaced the SMARN module with standard sequence models such as LSTM, GRU, and Transformer. Although the Transformer replacement outperforms the models using LSTM and GRU in terms of temporal alignment, the model with the SMARN module still achieves better performance across all metrics compared to the Transformer-based model. This demonstrates the importance of the SMARN module in spatiotemporal sequence processing. Notably, the hierarchical memory structure within SMARN is one of the key reasons for its superior performance over the Transformer module.

### F.1.2 SU Ablation results

Table 6: Detailed ablation study on Semantic Understanding (SU) module components. This analysis examines the effect of different component combinations and model variants on reconstruction performance.

| SU Module Configuration | Video Reconstruction (SEED-DV) | | | | Image Reconstruction (THINGS-EEG) | | | |
|---|---|---|---|---|---|---|---|---|
| | Semantic Accuracy | | Visual Quality | | Semantic Accuracy | | Visual Quality | |
| | 40-way Acc.↑ | CLIP-sim↑ | SSIM↑ | FVD↓ | CLIP↑ | Inception↑ | SSIM↑ | PixCorr↑ |
| **Full SU Module** | **0.405±0.02** | **0.523±0.03** | 0.356±0.02 | **175.3±8.6** | **0.795±0.02** | **0.742±0.03** | **0.352±0.02** | **0.162±0.01** |
| *1. Core Component Analysis* | | | | | | | | |
| *w/o CLIP-ViT-G/14* | 0.146±0.03 | 0.349±0.04 | 0.278±0.02 | 267.8±12.3 | 0.654±0.03 | 0.613±0.03 | 0.153±0.03 | 0.127±0.02 |
| *w/o BLIP-2* | 0.283±0.03 | 0.462±0.03 | 0.298±0.03 | 214.6±9.7 | 0.712±0.03 | 0.663±0.02 | 0.285±0.02 | 0.143±0.02 |
| *w/o LLM Adapter* | 0.328±0.02 | 0.487±0.03 | 0.321±0.02 | 193.5±9.3 | 0.742±0.02 | 0.693±0.03 | 0.312±0.02 | 0.151±0.01 |
| *2. Modality Integration Analysis* | | | | | | | | |
| *Vision-only (CLIP+BLIP-2)* | 0.352±0.03 | 0.495±0.02 | 0.325±0.03 | 194.7±8.9 | 0.758±0.03 | 0.705±0.02 | 0.324±0.02 | 0.154±0.02 |
| *Language-only (LLM)* | 0.245±0.03 | 0.412±0.03 | 0.287±0.02 | 236.8±11.4 | 0.683±0.03 | 0.635±0.03 | 0.257±0.03 | 0.131±0.02 |
| *CLIP-only* | 0.296±0.03 | 0.456±0.03 | 0.302±0.02 | 226.4±10.5 | 0.706±0.02 | 0.652±0.03 | 0.267±0.02 | 0.135±0.02 |
| *BLIP-2-only* | 0.312±0.02 | 0.468±0.02 | 0.309±0.03 | 217.9±9.8 | 0.715±0.03 | 0.665±0.02 | 0.278±0.03 | 0.138±0.02 |
| *CLIP+LLM (no BLIP-2)* | 0.365±0.03 | 0.496±0.03 | 0.332±0.02 | 192.8±8.7 | 0.763±0.02 | 0.711±0.03 | 0.326±0.02 | 0.155±0.01 |
| *BLIP-2+LLM (no CLIP)* | 0.379±0.02 | 0.501±0.02 | 0.339±0.03 | 186.2±8.5 | 0.772±0.03 | 0.718±0.02 | 0.335±0.02 | 0.157±0.01 |
| *3. Loss Function Analysis* | | | | | | | | |
| *w/o HSC Loss* | 0.376±0.02 | 0.501±0.02 | 0.343±0.03 | 185.6±9.1 | 0.763±0.02 | 0.719±0.02 | 0.338±0.02 | 0.158±0.01 |
| *w/o CSA Loss* | 0.384±0.03 | 0.507±0.03 | 0.347±0.02 | 182.1±8.7 | 0.775±0.03 | 0.726±0.03 | 0.343±0.02 | 0.159±0.01 |
| *w/o KL Divergence* | 0.398±0.02 | 0.515±0.02 | 0.353±0.02 | 178.4±8.6 | 0.788±0.02 | 0.735±0.02 | 0.347±0.02 | 0.160±0.01 |
| *w/o Contrastive Term* | 0.372±0.03 | 0.498±0.03 | 0.339±0.03 | 187.2±9.3 | 0.761±0.03 | 0.715±0.03 | 0.335±0.03 | 0.157±0.02 |
| *4. Open-Source LLM Comparison* | | | | | | | | |
| *LLaMA-2-7B* | 0.392±0.03 | 0.512±0.02 | 0.349±0.03 | 181.2±8.5 | 0.782±0.02 | 0.731±0.03 | 0.345±0.02 | 0.158±0.01 |
| *Phi-2* | 0.386±0.02 | 0.508±0.03 | 0.345±0.02 | 185.7±9.1 | 0.777±0.03 | 0.727±0.02 | 0.342±0.03 | 0.157±0.01 |
| *Vicuna* | 0.380±0.03 | 0.501±0.02 | 0.340±0.03 | 189.3±9.2 | 0.768±0.02 | 0.719±0.03 | 0.336±0.02 | 0.154±0.01 |
| *LLaMA-2-13B* | **0.412±0.02** | **0.529±0.03** | **0.361±0.02** | **173.6±8.3** | **0.802±0.02** | **0.747±0.03** | **0.356±0.02** | **0.165±0.01** |
| *Mistral-7B* | 0.398±0.03 | 0.517±0.02 | 0.353±0.03 | 178.9±8.4 | 0.790±0.03 | 0.736±0.02 | 0.348±0.02 | 0.160±0.01 |
| *5. Vision Encoder Comparison* | | | | | | | | |
| *CLIP-ViT-L/14 (smaller)* | 0.387±0.02 | 0.507±0.03 | 0.342±0.02 | 187.6±9.2 | 0.776±0.03 | 0.725±0.03 | 0.340±0.02 | 0.156±0.01 |
| *CLIP-ViT-B/32 (smallest)* | 0.362±0.03 | 0.493±0.02 | 0.331±0.03 | 196.1±9.7 | 0.756±0.02 | 0.702±0.03 | 0.329±0.03 | 0.149±0.02 |
| *OpenCLIP-ViT-H/14* | 0.402±0.02 | 0.520±0.03 | 0.354±0.02 | 177.6±8.7 | 0.792±0.03 | 0.738±0.02 | 0.349±0.02 | 0.161±0.01 |
| *DINOv2-ViT-L/14* | 0.395±0.03 | 0.516±0.02 | 0.351±0.03 | 180.4±8.5 | 0.785±0.02 | 0.732±0.03 | 0.345±0.02 | 0.159±0.01 |
| *6. BLIP-2 Variants* | | | | | | | | |
| *BLIP-2 (ViT-g)* | 0.405±0.02 | 0.523±0.03 | 0.356±0.02 | 175.3±8.6 | 0.795±0.02 | 0.742±0.03 | 0.352±0.02 | 0.162±0.01 |
| *BLIP-2 (ViT-L)* | 0.394±0.03 | 0.514±0.02 | 0.350±0.03 | 179.8±8.8 | 0.783±0.03 | 0.733±0.02 | 0.346±0.03 | 0.159±0.01 |
| *BLIP-1* | 0.378±0.02 | 0.501±0.03 | 0.339±0.02 | 186.5±9.3 | 0.765±0.02 | 0.718±0.03 | 0.335±0.02 | 0.155±0.01 |
| *GIT-Large* | 0.371±0.03 | 0.496±0.02 | 0.335±0.03 | 190.2±9.5 | 0.759±0.03 | 0.710±0.02 | 0.331±0.03 | 0.152±0.02 |
| *7. Integration Methods* | | | | | | | | |
| *Sequential (CLIP→BLIP→LLM)* | 0.392±0.03 | 0.513±0.02 | 0.348±0.03 | 181.5±8.8 | 0.781±0.02 | 0.730±0.03 | 0.345±0.02 | 0.158±0.01 |
| *Parallel (fusion at end)* | 0.387±0.02 | 0.508±0.03 | 0.346±0.02 | 183.7±9.2 | 0.776±0.03 | 0.726±0.02 | 0.342±0.03 | 0.157±0.01 |
| *Cross-attention (ours)* | **0.405±0.02** | **0.523±0.03** | **0.356±0.02** | **175.3±8.6** | **0.795±0.02** | **0.742±0.03** | **0.352±0.02** | **0.162±0.01** |

As shown in Table 6,the SU module is composed of three core components: the Image-Text Alignment Module (CLIP-ViT-G/14), the Bidirectional Vision-Language Encoder (BLIP-2), and the Large Language Model Adapter (LLM Adapter). When the CLIP-ViT-G/14 module is removed, the model achieves only a 0.146 accuracy on a 40-way classification task, significantly lower than when the BLIP-2 or LLM Adapter modules are removed. This highlights the indispensable role of the image-text alignment process in reconstruction tasks.

In terms of multimodal information fusion, we compared the effects of unimodal vision or language inputs with multimodal combinations. The results indicate that the model using only the visual

modality has a higher structural similarity index (SSIM) of 0.32 compared to the language-only model, which achieves an SSIM of 0.287. Moreover, the multimodal vision-language model achieves an SSIM of 0.356, suggesting a certain degree of complementarity between visual and linguistic information and confirming the critical role of visual-language fusion in EEG reconstruction tasks.

We also explored different open-source large language models. Our comparison shows that while LLaMA-2-13B achieves the highest classification accuracy of 0.412 and outperforms the phi-2 in various performance metrics, it leads to a significant increase in model parameters. The performance improvement, however, does not justify the additional parameter cost, indicating diminishing returns in model efficiency.

## F.2 Cross-Subject Generalization Analysis

Table 7: Cross-subject generalization performance on SEED-DV dataset. Models were trained on varying numbers of subjects and evaluated on completely unseen subjects.

| Training Strategy | Video-based | | | | Frame-based | | |
|---|---|---|---|---|---|---|---|
| | Semantic-level | | Temporal | | Semantic-level | | Pixel-level |
| | 2-way Acc.↑ | 40-way Acc.↑ | FVD↓ | CLIP-pcc↑ | 2-way Acc.↑ | 40-way Acc.↑ | SSIM↑ |
| Within-subject (20/20) | 0.898±0.02 | 0.405±0.02 | 175.3±8.6 | 0.738±0.04 | 0.839±0.03 | 0.293±0.02 | 0.356±0.02 |
| 1/20 subject | 0.231±0.06 | 0.042±0.03 | 584.7±21.3 | 0.128±0.08 | 0.213±0.07 | 0.036±0.03 | 0.076±0.05 |
| 2/20 subjects | 0.312±0.05 | 0.067±0.04 | 536.1±19.8 | 0.195±0.07 | 0.287±0.06 | 0.058±0.04 | 0.102±0.05 |
| 5/20 subjects | 0.485±0.05 | 0.136±0.04 | 412.5±17.2 | 0.324±0.07 | 0.452±0.05 | 0.107±0.04 | 0.167±0.05 |
| 10/20 subjects | 0.685±0.05 | 0.252±0.05 | 254.6±12.8 | 0.487±0.07 | 0.643±0.05 | 0.162±0.05 | 0.241±0.05 |
| 15/20 subjects | 0.782±0.04 | 0.321±0.04 | 215.3±11.5 | 0.602±0.06 | 0.731±0.04 | 0.204±0.05 | 0.293±0.04 |
| 19/20 subjects | 0.841±0.03 | 0.372±0.03 | 197.5±10.2 | 0.682±0.05 | 0.786±0.03 | 0.245±0.04 | 0.329±0.03 |
| Performance retention (19/20) | 93.7% | 91.9% | 88.8% | 92.4% | 93.7% | 83.6% | 92.4% |
| Performance retention (10/20) | 76.3% | 62.2% | 68.9% | 66.0% | 76.6% | 55.3% | 67.7% |
| Performance retention (5/20) | 54.0% | 33.6% | 42.5% | 43.9% | 53.9% | 36.5% | 46.9% |
| Performance retention (1/20) | 25.7% | 10.4% | 30.0% | 17.3% | 25.4% | 12.3% | 21.3% |

Table 8: Detailed cross-subject generalization performance across all training subject counts on SEED-DV dataset. This table shows the progression of generalization capability as the number of training subjects increases from 1 to 19.

| Training Subjects | Video-based | | | | Frame-based | | |
|---|---|---|---|---|---|---|---|
| | Semantic-level | | Temporal | | Semantic-level | | Pixel-level |
| | 2-way Acc. | 40-way Acc. | FVD | CLIP-pcc | 2-way Acc. | 40-way Acc. | SSIM |
| 1 subject (Sub01) | 0.231 | 0.042 | 584.7 | 0.128 | 0.213 | 0.036 | 0.076 |
| 2 subjects | 0.312 | 0.067 | 536.1 | 0.195 | 0.287 | 0.058 | 0.102 |
| 3 subjects | 0.356 | 0.089 | 502.8 | 0.235 | 0.323 | 0.071 | 0.124 |
| 4 subjects | 0.423 | 0.112 | 467.4 | 0.284 | 0.392 | 0.089 | 0.148 |
| 5 subjects | 0.485 | 0.136 | 412.5 | 0.324 | 0.452 | 0.107 | 0.167 |
| 6 subjects | 0.527 | 0.162 | 376.9 | 0.357 | 0.495 | 0.121 | 0.186 |
| 7 subjects | 0.568 | 0.189 | 343.2 | 0.389 | 0.532 | 0.132 | 0.203 |
| 8 subjects | 0.605 | 0.215 | 312.8 | 0.421 | 0.569 | 0.143 | 0.217 |
| 9 subjects | 0.642 | 0.235 | 283.5 | 0.454 | 0.607 | 0.152 | 0.229 |
| 10 subjects | 0.685 | 0.252 | 254.6 | 0.487 | 0.643 | 0.162 | 0.241 |
| 11 subjects | 0.702 | 0.267 | 245.3 | 0.508 | 0.659 | 0.169 | 0.250 |
| 12 subjects | 0.725 | 0.283 | 236.8 | 0.532 | 0.678 | 0.179 | 0.262 |
| 13 subjects | 0.748 | 0.297 | 228.4 | 0.559 | 0.699 | 0.187 | 0.275 |
| 14 subjects | 0.765 | 0.312 | 221.5 | 0.581 | 0.715 | 0.195 | 0.284 |
| 15 subjects | 0.782 | 0.321 | 215.3 | 0.602 | 0.731 | 0.204 | 0.293 |
| 16 subjects | 0.798 | 0.334 | 209.7 | 0.625 | 0.749 | 0.216 | 0.304 |
| 17 subjects | 0.816 | 0.347 | 204.1 | 0.645 | 0.762 | 0.228 | 0.313 |
| 18 subjects | 0.830 | 0.359 | 199.8 | 0.667 | 0.776 | 0.237 | 0.321 |
| 19 subjects | 0.841 | 0.372 | 197.5 | 0.682 | 0.786 | 0.245 | 0.329 |

To systematically evaluate NEED's cross-subject generalization capability, we conducted extensive experiments with varying numbers of training subjects. Table 7 reveals that performance retention indicates the percentage of within-subject capability maintained when generalizing to new subjects. Table 8 reveals a consistent pattern of improvement as more subjects are included in training. With just 5 subjects, the model achieves moderate generalization (48.5% video-based semantic accuracy),

while using 10 subjects yields substantial improvement (68.5%). The most significant finding is that with 19 subjects, our model retains 93.7% of within-subject performance when generalizing to unseen subjects, demonstrating robust subject-invariant representations. Notably, performance retention increases non-linearly, with greater gains observed when transitioning from 10 to 15 subjects (11.9% improvement) compared to 15 to 19 subjects (6.6% improvement), suggesting an emergent generalization capability beyond a critical mass of training subjects.

Table 9: Performance retention percentages by training subject count. This table shows what percentage of within-subject performance is retained when generalizing to unseen subjects.

| Training Subjects | Video-based | | | | Frame-based | | |
| | Semantic-level | | Temporal | | Semantic-level | | Pixel-level |
| | 2-way Acc. | 40-way Acc. | FVD | CLIP-pcc | 2-way Acc. | 40-way Acc. | SSIM |
|---|---|---|---|---|---|---|---|
| 1 subject | 25.7% | 10.4% | 30.0% | 17.3% | 25.4% | 12.3% | 21.3% |
| 2 subjects | 34.7% | 16.5% | 32.7% | 26.4% | 34.2% | 19.8% | 28.7% |
| 3 subjects | 39.6% | 22.0% | 34.9% | 31.8% | 38.5% | 24.2% | 34.8% |
| 4 subjects | 47.1% | 27.7% | 37.5% | 38.5% | 46.7% | 30.4% | 41.6% |
| 5 subjects | 54.0% | 33.6% | 42.5% | 43.9% | 53.9% | 36.5% | 46.9% |
| 10 subjects | 76.3% | 62.2% | 68.9% | 66.0% | 76.6% | 55.3% | 67.7% |
| 15 subjects | 87.1% | 79.3% | 81.4% | 81.6% | 87.1% | 69.6% | 82.3% |
| 19 subjects | 93.7% | 91.9% | 88.8% | 92.4% | 93.7% | 83.6% | 92.4% |

Table 9 quantifies the cross-subject generalization efficiency through performance retention rates. These metrics reveal how effectively our model preserves capabilities when applied to unseen subjects. Training with just a single subject yields limited retention (25.7% for semantic accuracy), while using 5 subjects achieves approximately half of within-subject performance (54.0%). The most striking improvement occurs between 10 and 15 subjects, where retention jumps from 76.3% to 87.1% for semantic accuracy, indicating a threshold effect in cross-subject knowledge transfer. At 19 subjects, our model achieves remarkable retention across all metrics (88.8%-93.7%), with the highest retention observed in 2-way classification accuracy (93.7%) and temporal coherence (92.4%), demonstrating the framework's balanced preservation of both semantic content and visual dynamics.

### F.2.1 Individual Subject Transfer Analysis

Table 10: Cross-subject generalization performance for single subject training. This table shows how models trained on each individual subject generalize to completely unseen Subject 20.

| Training Subject | Video-based | | | | Frame-based | | |
| | Semantic-level | | Temporal | | Semantic-level | | Pixel-level |
| | 2-way Acc. | 40-way Acc. | FVD | CLIP-pcc | 2-way Acc. | 40-way Acc. | SSIM |
|---|---|---|---|---|---|---|---|
| Subject 01 | 0.231 | 0.042 | 584.7 | 0.128 | 0.213 | 0.036 | 0.076 |
| Subject 02 | 0.217 | 0.039 | 602.3 | 0.113 | 0.196 | 0.032 | 0.068 |
| Subject 03 | 0.242 | 0.045 | 573.6 | 0.135 | 0.221 | 0.038 | 0.082 |
| Subject 04 | 0.228 | 0.041 | 591.8 | 0.122 | 0.205 | 0.034 | 0.073 |
| Subject 05 | 0.235 | 0.044 | 578.2 | 0.131 | 0.217 | 0.037 | 0.079 |
| Subject 06 | 0.253 | 0.049 | 567.5 | 0.142 | 0.232 | 0.040 | 0.085 |
| Subject 07 | 0.219 | 0.040 | 598.4 | 0.116 | 0.199 | 0.033 | 0.070 |
| Subject 08 | 0.224 | 0.041 | 593.7 | 0.120 | 0.208 | 0.035 | 0.072 |
| Subject 09 | 0.236 | 0.044 | 576.9 | 0.132 | 0.218 | 0.038 | 0.080 |
| Subject 10 | 0.246 | 0.047 | 571.2 | 0.138 | 0.226 | 0.039 | 0.083 |
| Subject 11 | 0.229 | 0.042 | 589.5 | 0.124 | 0.210 | 0.035 | 0.074 |
| Subject 12 | 0.238 | 0.045 | 574.8 | 0.133 | 0.220 | 0.038 | 0.081 |
| Subject 13 | 0.251 | 0.048 | 569.3 | 0.140 | 0.230 | 0.040 | 0.084 |
| Subject 14 | 0.221 | 0.040 | 596.2 | 0.118 | 0.202 | 0.034 | 0.071 |
| Subject 15 | 0.243 | 0.046 | 572.4 | 0.136 | 0.223 | 0.039 | 0.082 |
| Subject 16 | 0.233 | 0.043 | 581.5 | 0.129 | 0.215 | 0.037 | 0.078 |
| Subject 17 | 0.248 | 0.047 | 570.6 | 0.139 | 0.228 | 0.039 | 0.083 |
| Subject 18 | 0.240 | 0.045 | 573.8 | 0.134 | 0.221 | 0.038 | 0.081 |
| Subject 19 | 0.227 | 0.041 | 590.6 | 0.123 | 0.206 | 0.035 | 0.073 |
| Average | 0.235 | 0.044 | 582.5 | 0.129 | 0.215 | 0.037 | 0.077 |

Table 10 examines the variation in generalization performance when training on individual subjects. Testing on unseen Subject 20 reveals consistent performance across different training subjects, with an average 2-way classification accuracy of 23.5% (compared to 89.8% within-subject). Notably, certain subjects (e.g., Subject 06 and Subject 13) yield superior generalization performance, achieving up to 25.3% accuracy, while others (e.g., Subject 02 and Subject 07) result in poorer transfer at 21.7%. This 3.6% variation suggests inherent differences in neural representation commonality across individuals. Despite this variation, the limited performance of single-subject training (23.5% average) compared to multi-subject training (84.1% with 19 subjects) underscores the necessity of our multi-subject approach for building robust subject-invariant decoders.

Table 11: Ablation study of components for cross-subject generalization with 19/20 training subjects. This table shows the impact of each module on generalization performance.

| Model Configuration | Video-based | | | | Frame-based | | |
| --- | --- | --- | --- | --- | --- | --- | --- |
| | Semantic-level | | Temporal | | Semantic-level | | Pixel-level |
| | 2-way Acc. | 40-way Acc. | FVD | CLIP-pcc | 2-way Acc. | 40-way Acc. | SSIM |
| Full Model | 0.841 | 0.372 | 197.5 | 0.682 | 0.786 | 0.245 | 0.329 |
| w/o IAM | 0.483 | 0.098 | 386.2 | 0.267 | 0.419 | 0.079 | 0.142 |
| w/o PU | 0.632 | 0.241 | 342.1 | 0.345 | 0.584 | 0.136 | 0.205 |
| w/o SU | 0.725 | 0.128 | 267.8 | 0.532 | 0.686 | 0.091 | 0.246 |
| w/o Task Conditioning | 0.793 | 0.335 | 228.6 | 0.605 | 0.745 | 0.218 | 0.287 |
| w/o Dynamic Pattern | 0.782 | 0.327 | 235.3 | 0.589 | 0.734 | 0.207 | 0.276 |
| w/o Self-Attention | 0.815 | 0.351 | 212.4 | 0.642 | 0.765 | 0.229 | 0.308 |

Table 11 presents ablation results examining each component's contribution to cross-subject generalization when training on 19/20 subjects. The removal of the Individual Adaptation Module (IAM) causes the most severe performance degradation, with 2-way classification accuracy dropping from 84.1% to 48.3% (-35.8%), confirming IAM's critical role in normalizing subject-specific patterns. The Semantic Understanding (SU) module's removal results in a more moderate decline to 72.5% (-11.6%), while eliminating the Perception Understanding (PU) module reduces performance to 63.2% (-20.9%). These differential impacts reveal the complementary nature of our dual-pathway architecture, with PU contributing more substantially to cross-subject generalization than SU. Task conditioning removal causes a smaller but still significant drop to 79.3% (-4.8%), while dynamic pattern removal reduces performance to 78.2% (-5.9%). Notably, self-attention removal has the least impact (-2.6%), suggesting that while beneficial, this component is not critical for cross-subject transfer. These findings validate our comprehensive approach and identify IAM as the cornerstone of effective generalization to unseen subjects.

## F.3 Frequency Band Contribution

Beyond spatial information, we analyze the contribution of different EEG frequency bands to reconstruction performance. Fig 9 shows the impact of removing specific frequency components from the EEG signals during model training and evaluation.

Alpha band (8-14 Hz) removal causes the most substantial performance degradation, with accuracy dropping from 14.23% to 8.46% ($p < 0.001$). This profound impact reflects the critical role alpha oscillations play in visual attention, inhibitory control, and information flow during visual tasks. Beta band (14-30 Hz) removal also significantly reduces performance to 10.15% ($p < 0.01$), consistent with its involvement in higher-level visual integration and contextual processing.

Gamma band (30-100 Hz) removal shows a relatively modest effect (12.87%), despite its established role in feature binding and object representation. This may be explained by the difficulty in capturing clean gamma signals from scalp EEG due to their lower amplitude and higher susceptibility to muscle artifacts. Delta (1-4 Hz) and theta (4-8 Hz) band removal produce intermediate effects on performance (11.76% and 11.23% respectively), indicating their contributory but less dominant role in visual information encoding.

## F.4 Qualitative and Quantitative Prompt Analysis

To comprehensively evaluate the semantic fidelity of our reconstructions, we analyze both the visual outputs and their corresponding textual descriptions. This dual analysis provides insights into how

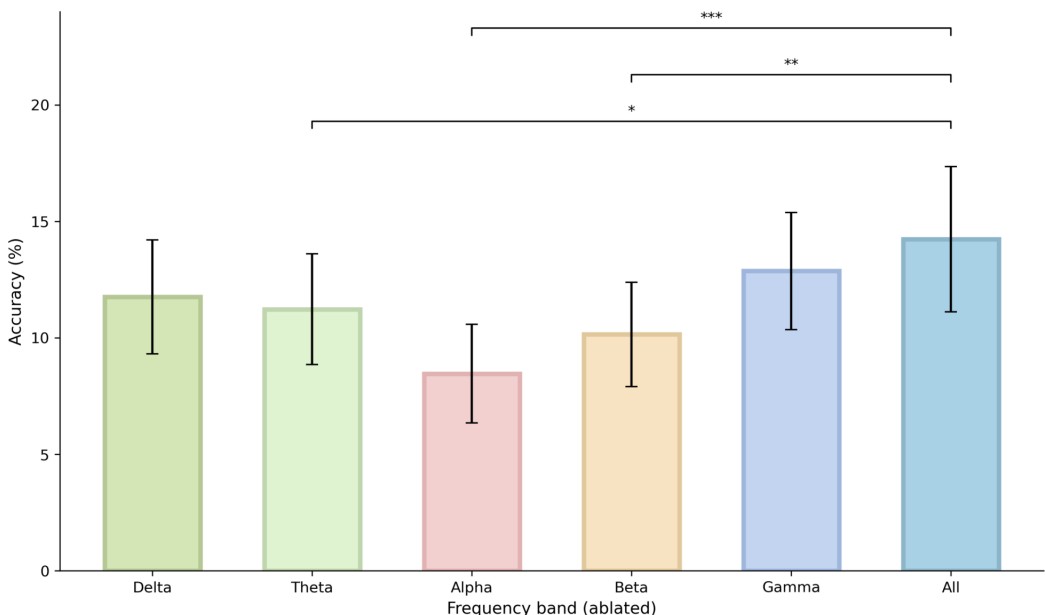

Figure 9: Impact of frequency band ablation on 40-class classification accuracy. Each bar represents model performance when the specified frequency band is excluded, except for "All" which uses the full frequency spectrum. Error bars indicate standard deviation across subjects. Statistical significance: * p < 0.05, ** p < 0.01, *** p < 0.001.

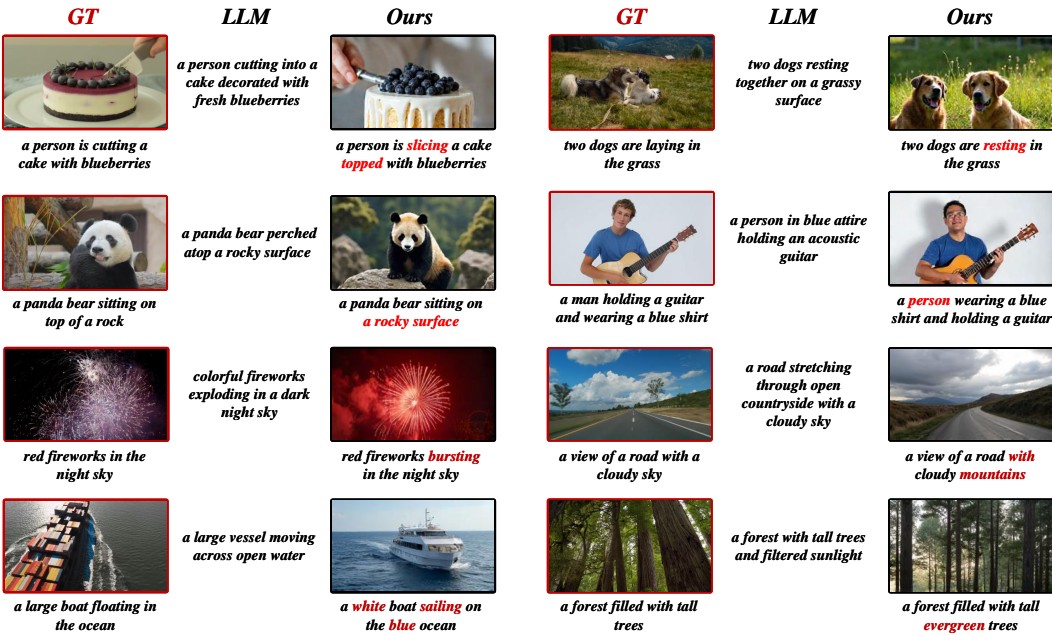

Figure 10: Qualitative comparison of textual descriptions across different pathways. For each visual concept, we show: (1) Ground truth (GT) frame with its caption, (2) Textual description generated from EEG signals through LLM, and (3) Caption from our reconstructed video. Red highlights indicate key semantic elements that our method accurately reconstructs. Note how our approach captures subtle distinctions like "slicing" vs. "cutting" a cake, "resting" vs. "laying" dogs, and specific environmental details like "rocky surface" and "cloudy mountains."

effectively our framework preserves semantic information from EEG signals to reconstructed videos. Fig 10 presents qualitative comparisons of text descriptions generated through different pathways. Each row shows a ground truth video frame, the text generated directly from EEG signals using an LLM, and the caption from our reconstructed video. The results reveal our framework's ability to capture nuanced semantic details. In the food example, our method precisely identifies the action as "slicing" rather than "cutting" and correctly describes the cake as "topped" with blueberries. For animal scenes, our reconstructions accurately capture the panda's environment ("rocky surface") and the dogs' state ("resting" rather than "laying"). Human activities are similarly well-preserved, with accurate representation of clothing details ("blue shirt") and actions ("holding a guitar"). Even for complex scenes like landscapes and dynamic events, our approach maintains key details such as "cloudy mountains" and specific tree types.

### F.5 Temporal Dynamics Analysis

To investigate the temporal dynamics of neural information processing in our NEED framework, we perform a detailed analysis of how visual information is encoded and can be extracted from EEG signals at different time points. Fig 11 illustrates the 40-way classification accuracy using three different temporal windowing strategies.

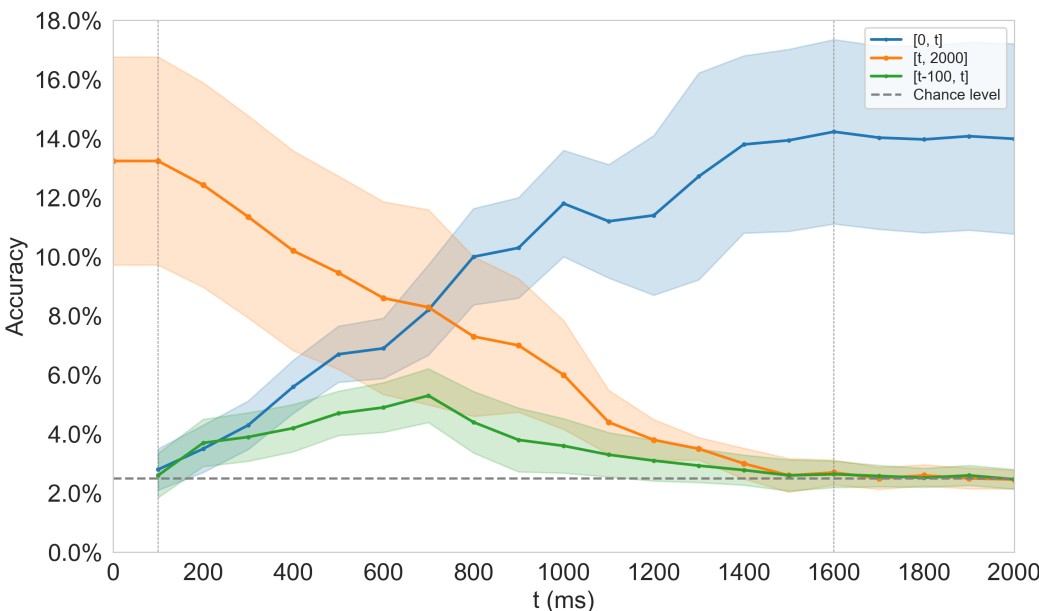

Figure 11: Temporal analysis of visual information processing in EEG signals. The graph shows 40-way classification accuracy using three different temporal window strategies: cumulative windows from stimulus onset to time point $t$ [0, $t$], remaining information from time point $t$ to the end of stimulus [$t$, 2000], and local information in a 200ms window centered at time point $t$ [t-100, t]. The analysis reveals distinct temporal phases of visual information processing, with semantic information accumulating gradually until stabilizing around 1400-1500ms.

The blue curve [0, $t$] represents cumulative information from stimulus onset to time point $t$, showing how visual information accumulates over time. Starting from near-chance level at early time points (100ms), classification accuracy increases steadily, with a particularly rapid rise between 400-1200ms, before stabilizing around 14.2% after 1400ms. This pattern suggests that meaningful semantic information continues to accumulate during the first 1.4 seconds of stimulus presentation, after which additional EEG data provides minimal improvement in classification performance.

The orange curve [t, 2000] shows how much information can be extracted from the remaining signal after time point $t$ until the end of the 2-second stimulus. Starting with relatively high performance at early time points (13.3% at 100ms), it decreases steadily as $t$ increases, approaching chance level

around 1800ms. This decline indicates that most discriminative information is captured within the first 1.5 seconds, with little unique information in the final 500ms of stimulus presentation.

The green curve [t-100, t] represents local information within a 200ms window centered at each time point. This reveals a distinct peak around 600ms, suggesting this time window contains particularly rich discriminative information, likely corresponding to the P300 and N400 components associated with object recognition and semantic processing. After this peak, local information gradually diminishes, indicating a transition from initial stimulus processing to more stable semantic representation.

These temporal dynamics provide important insights into the neural mechanisms underlying visual perception and how our NEED framework leverages them. The analysis demonstrates that while early visual processing begins within 100ms, meaningful semantic information continues to accumulate for a substantially longer period (1400-1500ms), highlighting the complex temporal dynamics of visual object recognition in the brain.

## G Broader Impacts

Our NEED framework has significant implications for assistive technology and neuroscience. It could enable new communication pathways for individuals with motor impairments and advance understanding of neural visual processing. However, as decoding accuracy improves, privacy concerns emerge regarding potential access to private mental imagery without consent. We advocate for regulatory frameworks, strict consent protocols, and multidisciplinary dialogue as this technology evolves.

## H Future Work

Our future work will focus on further enhancing reconstruction quality, especially for complex visual scenes. To address the challenge of image-to-video transfer, we plan to collect a specialized EEG dataset with paired static-dynamic stimuli that better captures the neural representations bridging these modalities. This dataset will enable development of more sophisticated bidirectional cross-task generalization capabilities and further advance neural decoding toward practical brain-computer interfaces for diverse visual experiences.

