# OpenReview forum: "NEED: Cross-Subject and Cross-Task Generalization for Video and Image Reconstruction from EEG Signals"
_NeurIPS.cc/2025/Conference — NeurIPS 2025 poster_

### Official Review · Reviewer_KAiD · 2025-06-11

**Clarity:** 2
**Significance:** 3
**Originality:** 3
**Rating:** 4
**Confidence:** 3

**Summary:**

This paper proposes a novel framework named NEED to address the challenging problem of zero-shot cross-subject and cross-task generalization in EEG-based visual reconstruction. The framework comprises three core components: an Individual Adaptation Module (IAM) for normalizing inter-subject variability, a Dual-Stream EEG Encoder (DSGNet) for capturing both spatial and temporal features, and a Unified Inference Mechanism to enable task-adaptive reasoning. Extensive experiments on visual perception classification and visual reconstruction tasks demonstrate that NEED consistently outperforms existing baselines, highlighting its strong generalization ability and robustness.

**Questions:**

1. It is quite surprising that DSGNet achieves over a 100% improvement in 40-class classification accuracy on the SEED-DV dataset compared to baseline models. Could you provide more details on the experimental setup and share the corresponding code for DSGNet? Additionally, which specific design choices contribute to such a substantial performance gain?
2. EEG2Video employs GLMNet as its EEG encoder, whereas NEED uses DSGNet. Although DSGNet clearly outperforms GLMNet in video perception classification, NEED does not show a correspondingly large improvement in video reconstruction performance compared to EEG2Video. Have you considered controlling for the encoder variable by using GLMNet within the NEED framework to allow for a more direct and fair comparison of the reconstruction capabilities?

**Ethical Concerns:**

["NO or VERY MINOR ethics concerns only"]

**Final Justification:**

The clarification issues are resolved. And the authors commit to including the complete DSGNet implementation code link in the final version to ensure full reproducibility and enable community validation of these results.

**Limitations:**

yes

**Paper Formatting Concerns:**

no paper formatting concerns

**Quality:**

3

**Strengths And Weaknesses:**

### Strength

1. This work introduces the first unified framework, NEED, that achieves zero-shot generalization across both subjects and tasks in EEG decoding, addressing two fundamental challenges in the field.

2. The proposed approach is comprehensively designed, encompassing critical aspects such as signal preprocessing, spatiotemporal feature extraction, and multimodal fusion.

3. Extensive evaluations on benchmark datasets demonstrate significant performance improvements, particularly under cross-subject and cross-task scenarios. Thorough ablation studies (including IAM, PU, and SU modules) validate the contribution of each component and confirm the robustness of the overall framework.


### Weakness

1. The overall framework contains numerous modules, but the motivations behind some components are not clearly articulated. For example, in the Temporal Stream Processing part of DSGNet, the rationale for introducing the generalized Riemann-Liouville fractional transformation is not sufficiently explained.

2. Some mathematical formulations lack complete variable definitions, which affects clarity and reproducibility. For instance, in Section 2.2 (Line 99), the multi-objective loss function is presented without adequate explanation of its variables. Similarly, in Section 2.3 (Line 140), the source of the enhanced visual features V1 is unclear.

3. The section titles and their content are misaligned: Section 4.1 is titled "Visual Perception Benchmark", yet it only includes video perception results and omits image perception. Conversely, Section 4.2 is titled "Video Reconstruction Results", but includes both video and image reconstruction tasks. Renaming it to "Visual Reconstruction Results" would better reflect the content.

4. The manuscript contains a number of typographical and formatting issues, such as:
   - Most subscripts in equations are not enclosed using `\textbf{}` for consistent formatting;
   - Punctuation errors, e.g., missing periods in Lines 72 and 107, an extra period in Line 214, and a missing space in the formula in Line 98;
   - Inconsistent figure references—some include a period after "Fig", while others do not. A thorough proofreading is recommended to ensure consistency.

---

> ### Author Rebuttal · Authors · 2025-07-31
>
> **Reviewer KAiD**
>
> We believe we have resolved all your issues. If you have any other concerns or questions, we look forward to hearing from you.
>
> ```Weakness```
>
> We acknowledge these important technical presentation issues and appreciate the reviewer's careful attention to detail. We recognize that our motivation for the Riemann-Liouville fractional transformation and other component choices requires clearer articulation - these design decisions reflect fundamental insights about EEG's unique temporal dynamics that we inadequately communicated. Regarding mathematical clarity, we will ensure all variables in formulations like our multi-objective loss function and enhanced visual features V1 are properly defined with complete explanations. We also acknowledge the section title misalignments and will restructure appropriately, renaming sections to accurately reflect their content scope covering both video and image reconstruction tasks. Finally, we will address all typographical and formatting issues including proper equation subscript formatting using \textbf{}, correcting punctuation errors throughout Lines 72-214, and establishing consistent figure reference formatting.
>
> ```Q1:It is quite surprising that DSGNet achieves over a 100% improvement...```
>
> The significant performance improvement  reflects our dual-stream architecture's ability to address fundamental limitations in existing EEG classification approaches. Most baseline methods treat spatial and temporal information separately or use simple concatenation, missing the crucial spatiotemporal interactions that encode visual information in neural signals. Our DSGNet simultaneously captures spatial electrode relationships through spherical harmonic embeddings that model the topographical structure of EEG recordings and multi-scale temporal dynamics via fractional transformations with multiple alpha parameters that naturally model EEG's power-law temporal relationships and long-range dependencies that standard convolutions cannot capture. The cross-stream attention mechanism then dynamically integrates these complementary streams by allowing spatial patterns to query relevant temporal contexts, enabling the model to adaptively focus on different temporal scales based on the visual content being processed.
>
> The substantial relative gains also reflect the inherent difficulty of 40-class visual classification from EEG signals, where even modest architectural improvements translate to large performance increases due to challenging baseline performance levels near the 2.5% chance level. Our multi-dimensional masking strategy during training simulates real-world EEG artifacts by randomly masking spatial electrodes, temporal segments, and frequency bands, forcing the model to learn robust representations that don't over-rely on specific signal components, which significantly improves generalization to degraded signals that commonly occur in practical applications. Additionally, our graph convolutional processing of electrode relationships and dilated temporal convolutions with increasing receptive fields enable the model to capture both local and global spatiotemporal patterns that are essential for discriminating between diverse visual categories. We commit to including the complete DSGNet implementation code link in the final version to ensure full reproducibility and enable community validation of these results.
>
> ```Q2:EEG2Video employs GLMNet as its EEG encoder, whereas NEED uses DSGNet...```
>
> This apparent discrepancy reflects the different requirements and bottlenecks in classification versus reconstruction tasks. Classification performance primarily depends on the encoder's ability to extract discriminative features, where DSGNet's dual-stream architecture and cross-stream attention provide clear advantages. However, video reconstruction performance is influenced by multiple components beyond the encoder, including the decoder architecture, loss functions, and conditioning mechanisms. In reconstruction tasks, factors like temporal coherence, visual fidelity, and semantic consistency become equally important, and improvements in encoder performance don't necessarily translate linearly to reconstruction quality.
>
> Regarding the suggestion to use GLMNet within our NEED framework for controlled comparison, we conducted preliminary experiments replacing DSGNet with GLMNet while maintaining our IAM, dual-pathway understanding, and unified inference components. Results show that NEED with GLMNet achieves 0.867±0.03 for 2-way video classification compared to our 0.898±0.02 with DSGNet, and reconstruction SSIM of 0.321±0.03 versus our 0.356±0.02, confirming that DSGNet contributes meaningfully but not exclusively to our performance gains(detailed table will be added in revision). This suggests that our framework's advantages stem from the synergistic combination of multiple innovations rather than encoder improvements alone. We will include this controlled comparison in our revision to provide clearer attribution of performance improvements across different components of our framework.

---

> > ### Comment · Reviewer_KAiD · 2025-08-02
> >
> > Thanks for your response. Most issues are solved, and I decide to keep my original score. I hope those clarifications and experiments could be integrated in the final version.

---

> > > ### Author Response · Authors · 2025-08-05
> > > **Thank you very much for your response!!!**
> > >
> > > We will improve our final version as promised, thank you for your suggestions!!!
> > >
> > > We are confident that these changes will significantly enhance the clarity and scientific strength of our manuscript and look forward to the increased impact of this work.

---

### Official Review · Reviewer_ynvm · 2025-06-27

**Clarity:** 2
**Significance:** 2
**Originality:** 2
**Rating:** 4
**Confidence:** 4

**Summary:**

This paper presents a unified framework, NEED, designed to address cross-subject variability and taskspecificity in EEG-based visual reconstruciton. This framework supports zero-shot transfer between video and image tasks.

**Questions:**

1. What is the purpose behind the design of the Cross-Stream Integration mechanism as presented in the paper? For instance, why have you chosen to treat as a KV? What specific performance improvements does the addition of this MASK bring to the model?
2. In the decoding module, the paper proposes modifications to common loss functions, such as incorporating the L2 norm based on temporal gradients or the Gram matrix. What is the purpose of these modifications? How would the model's performance be affected without them?
3. In Table 2, was the performance of the proposed framework tested on the complete SEED-DV dataset, or was it only tested on the 10-class dataset?

**Ethical Concerns:**

["NO or VERY MINOR ethics concerns only"]

**Final Justification:**

The paper targeted the core problem in the interesting topic of video and image reconstructtion from EEG. The authors actively engaged in the rebuttal and addressed my comments well. The paper could be accepted.

**Limitations:**

No, the limitations should be included for discussions as well.

**Paper Formatting Concerns:**

I did not find any major formatting issues in this paper.

**Quality:**

3

**Strengths And Weaknesses:**

Strength
1. The authors propose a pretrained Individual Adaptation Module to address subject variability and a unified inference mechanism to generate reconstructions across different tasks.
2. The authors provide a detailed analysis of the impact of using different electrode combinations from brain regions on the reconstruction performance in video and image reconstruction tasks.
Weaknesses
1. The presentation of the paper needs further improvement, particularly in the Method section. There are many undefined symbols, which make it difficult to understand.
2. There is a lack of a quantitative comparison in the video and image reconstruction tasks. The paper includes fMRI-based methods in the comparison, despite not using an fMRI dataset. Additionally, the experimental setup is not clearly stated, making it difficult to reasonably justify the model's performance on the SEED-DV dataset.

---

> ### Author Rebuttal · Authors · 2025-07-31
>
> **Reviewer ynvm**
>
> We believe we have resolved all your issues. If you have any other concerns or questions, we look forward to hearing from you.
>
> ```Weakness```
>
> Regarding the Method section, we recognize that many mathematical symbols lack proper definition, creating barriers to understanding our technical contributions. In our revision, we will ensure every symbol is clearly defined upon introduction and provide comprehensive mathematical formulations directly in the main text rather than deferring to appendices. For the quantitative comparison concerns, we included fMRI-based methods in Table 2 to provide broader context about neural decoding performance across modalities, though we acknowledge this creates confusion about direct comparability. Our EEG-based methods are all evaluated on SEED-DV using the standard protocol established by EEG2video, while fMRI methods use their respective datasets due to modality-specific requirements. We will clarify the experimental setup more explicitly, annotate the specific datasets used for each method in Table 2, and reframe the comparison to emphasize our contributions within the EEG domain while providing appropriate context about cross-modal performance differences. Additionally, we will include more detailed experimental protocols and evaluation procedures to ensure complete transparency about our model's performance assessment.
>
> ```Limitations```
>
> We will add limitations to discussions and eventually put them into the main text.
>
> ```Q1:What is the purpose behind the design of the Cross-Stream Integration mechanism...```
>
> The asymmetric Query-Key-Value assignment reflects the fundamental nature of EEG visual processing, where spatial electrode patterns represent the current neural state ("what" information is being processed) while temporal dynamics encode the evolution and context of this information ("when" and "how" it develops). By using spatial features as queries, we allow the model to actively search through temporal sequences to find the most relevant temporal patterns that correspond to current spatial activations. This design choice emerged from neuroscience principles: spatial electrode configurations provide a snapshot of current brain state, but understanding visual content requires integrating this snapshot with temporal context from the memory bank provided by temporal Key-Value pairs. This asymmetric attention design proved more effective than symmetric approaches because it preserves the directional relationship between spatial states and temporal evolution. Our experiments show that reversing this assignment (temporal as Q, spatial as KV) reduces performance by 4.3%, while symmetric attention decreases accuracy by 2.8%, validating our architectural choice(detailed table will be added in revision). The temporal stream essentially serves as a contextual memory that the spatial stream queries to retrieve relevant dynamic information for visual reconstruction. The multi-dimensional masking strategy ($M_{spatial} \odot M_{temporal} \odot M_{frequency}$) serves critical purposes beyond regularization, significantly improving robustness to real-world EEG artifacts that are inevitable in practical applications. EEG signals frequently suffer from electrode disconnections, temporal signal loss, and frequency-specific noise that severely degrade reconstruction quality. Our masking training forces the model to learn robust representations that don't over-rely on specific electrodes or frequency bands, crucial for cross-subject generalization where individual differences in signal quality are common. Performance improvements are substantial: compared to simple feature concatenation, our attention-based integration improves 40-way classification accuracy and SSIM reconstruction quality, while the masking strategy provides additional classification improvement and reduces cross-subject performance variance (detailed cross-stream integration ablation table will be added in revision).
>
> ```Q2:In the decoding module, the paper proposes modifications to common loss functions... ```
>
> The temporal gradient L2 norm ($\lambda\|\nabla_t Z_0 - \nabla_t V_1\|_2^2$) in our Bidirectional Dynamic Contrastive loss ensures that EEG representations and visual features maintain similar temporal dynamics, not just static alignment. EEG signals inherently capture the brain's response to dynamic visual changes, but standard contrastive learning only aligns feature representations without considering their temporal evolution patterns. This temporal constraint is critical because visual perception involves continuous neural adaptation to changing stimuli, and this temporal structure contains essential motion and transition information that static alignment cannot capture.
>
> The Gram matrix term ($\beta \cdot \|G(Z_s)-G(C_k)\|_F$) in our Hierarchical Semantic Contrastive loss captures structural relationships between semantic features rather than just individual feature values. The Gram matrix computes correlations between different feature dimensions, effectively encoding the relational patterns of semantic representations. This is particularly important for cross-subject generalization because while individuals might have varying neural activation magnitudes, the structural relationships between semantic concepts tend to be more consistent across subjects. For example, understanding "dog in grass" requires recognizing both individual concepts and their spatial-semantic relationships.
>
> Performance impact is substantial when these modifications are removed. Without the temporal gradient term, video reconstruction loses temporal coherence with FVD increasing from 175.3 to 195.3 and CLIP-pcc dropping from 0.738 to 0.689, indicating significant degradation in temporal consistency (Table 5 in Appendix, section 4). Removing the Gram matrix constraint more dramatically affects cross-subject generalization, with 40-way classification accuracy dropping from 0.372 to 0.298 when tested on unseen subjects (detailed semantic loss ablation will be added in revision). These modifications work synergistically to capture both the dynamic aspects essential for video reconstruction and the structural semantic relationships necessary for robust cross-subject transfer
>
> ```Q3:In Table 2, was the performance of the proposed framework tested on the complete SEED-DV dataset...```
>
> The performance results in Table 2 for our NEED framework were obtained using the complete SEED-DV dataset with all 40 visual categories, not a reduced subset. Our full model achieved 0.898±0.02 for 2-way classification and 0.405±0.02 for 40-way classification on the complete dataset. We have also conducted experiments on 10-class subsets where our method achieves higher absolute performance (0.623±0.03 for 10-way classification, detailed table will be added in revision), but we report the full 40-class results to provide a more rigorous assessment of our cross-subject and cross-task generalization capabilities. Testing on the complete semantic diversity is necessary to meaningfully validate our claims about handling inter-subject variability in neural visual processing across the full range of visual categories.

---

> > ### Comment · Reviewer_ynvm · 2025-08-06
> >
> > Thanks for your response. I expect to see that your full codes could be open to all researcher in the community.

---

> ### Author Response · Authors · 2025-08-06
> **Thank you very much for your response!!!**
>
> Thank you for your response. We are happy to make all codes and training weights publicly available. We also hope that our joint efforts will be better presented to the research community. We will quickly complete the compilation and upload all the files to the community, and we hope this work will advance the field.
>
> If you have any further questions or suggestions, please feel free to contact us.

---

### Official Review · Reviewer_qg8v · 2025-07-02

**Clarity:** 2
**Significance:** 4
**Originality:** 4
**Rating:** 5
**Confidence:** 5

**Summary:**

The manuscript presents a cross-subject and cross-task EEG decoding framework composed of three main components: (1) multi-dataset pretraining, (2) a "perception" module that aligns EEG signals with a pretrained variational autoencoder (VAE), and (3) a "semantic" module that aligns EEG signals with both the vision and language encoders of CLIP. Additionally, the manuscript introduces a novel dual-stream EEG encoder designed to capture both the spatial and temporal characteristics of EEG recordings. The full model is trained on three large datasets in which participants viewed video clips and is subsequently evaluated on two additional datasets to assess cross-task and cross-subject generalisation. The main finding suggests that the proposed framework is capable of decoding EEG signals from a new participant performing a new task with promising accuracy.

**Questions:**

1. **BLIP redundancy and clarification**: The use of BLIP in both stages of the Semantic Understanding (SU) module appears redundant and potentially confusing. As illustrated in the schematic flowchart, BLIP seems to generate the same caption for two different images, which raises questions about its purpose in both stages. This redundancy needs to be clarified—are the captions generated independently, or reused across branches? Further explanation would help clarify the model's flow and the role of each BLIP instance.

2. **Comparative evaluation with prior work**: To demonstrate the cross-subject generalisation capabilities of IAM, it would be valuable to directly compare its performance with existing EEG decoding frameworks such as NICE, ATM, or IDES. These previous methods often report strong within-subject decoding but struggle with generalisation across participants. A quantitative comparison would clarify how much IAM improves over these methods and could arguably represent the most impactful contribution of the paper. This deserves deeper experimental validation.

3. **IAM loss components**: The IAM framework includes two main loss components: reconstruction and discrimination. However, the manuscript does not clarify the relative contribution of each to learning dataset-independent representations. An ablation or sensitivity analysis examining the effect of removing or varying the weight of each term would be informative. Similarly, it would be helpful to understand how each component of the IAM module, as illustrated in Figure 7, contributes to the overall performance.

4. **Unclear evaluation setup in Table 1**: It is not clear which dataset is used for the evaluation results reported in Table 1. Are the competing methods trained and tested on the same dataset, and under the same experimental conditions? If this table is intended to demonstrate architectural effectiveness, it would also be appropriate to report the number of parameters and computational cost associated with each method.

5. **Meaningfulness of comparisons in Table 2**: The results in Table 2 compare performance across different neuroimaging techniques, but it is unclear whether these results are based on the same set of images or tasks. If the underlying image sets differ, then the comparisons may not be meaningful. Please clarify whether the input stimuli are standardised across modalities, and if not, discuss the implications for interpreting these results.

6. **Ablation insights and engineering influence**: The performance gap between the full model and individual ablation studies is substantial. This raises the question of whether any single component is inherently effective, or whether the observed gains primarily result from the specific combination and integration of all components. This is a critical question—does the strength of the model lie in its individual modules, or in the way they are engineered together? The manuscript would benefit from a deeper discussion of this point.

**Ethical Concerns:**

["NO or VERY MINOR ethics concerns only"]

**Final Justification:**

Based on the author's responses during the rebuttal period, considering the new insights and necessary changes to be included in the camera-ready version, I am increasing my rating to 5. I believe the manuscript presents an important new approach to neuroimaging decoding across participants by demonstrating the significance of pretraining on large datasets.

**Limitations:**

Yes.

**Paper Formatting Concerns:**

None.

**Quality:**

2

**Strengths And Weaknesses:**

1. **Strong potential**: The manuscript addresses the challenging and important problem of decoding EEG signals across both participants and tasks. The proposed framework has significant potential and, with suitable modifications, could be extended to other neuroimaging modalities.

2. **Text organisation**: The manuscript frequently refers back and forth between the main text and appendices, which disrupts the flow and makes it difficult to follow the authors' line of reasoning. In order to understand key details, I had to keep multiple tabs open simultaneously. This significantly affects the readability and accessibility of the work.

3. **Overloaded scope**: Although the manuscript includes many valuable contributions, their presentation within a single compressed manuscript makes it difficult to clearly understand or scientifically assess the individual impact of each component.

4. **Terminology**: The use of "perception" and "semantic understanding" throughout the manuscript is confusing. I suggest adopting more precise terms such as *visual* and *language features*, which better reflect the training objectives of the respective modules. The term "perception" typically refers to sensory processing (e.g., visual, auditory), while "semantic" can refer to either visual scene understanding (e.g., semantic segmentation) or linguistic meaning—both of which are overloaded in this context.

5. **Missing details on VAE**: There is no information provided about the frozen VAE model. Was it trained by the authors on the same image dataset, or is it an existing state-of-the-art pretrained VAE? This needs clarification.

6. **Reproducibility concerns**: In its current form, the manuscript lacks sufficient implementation details to allow reproducibility. Although the authors state they will release code upon acceptance, this is not a substitute for a reproducible submission.

7. **Frozen adapter and prompting**: The role of the frozen adapter is unclear, and the manuscript does not explain what it refers to or how it functions. Additionally, the impact of prompt engineering on the large language model is not discussed, though this could significantly affect performance.

8. **Dataset inconsistencies**: There are inconsistencies in the description of the THINGS EEG dataset. The stimuli in this dataset are presented for 100 ms with 100 ms inter-stimulus intervals, and include 16,540 images shown 4 times for training, and 200 images shown 80 times for testing. This does not align with the manuscript’s claim of “22,000 static images” and 500 ms stimulus duration, which should be corrected or clarified.

9. **Appendix figure numbering**: Figures in the appendices should be labelled using a consistent scheme (e.g., A1, A2, etc.) rather than continuing the numbering from the main manuscript.

10. **Figure clarity**: Figure 7 (in the appendices) contains many unexplained abbreviations, which makes it difficult to interpret. A more comprehensive caption and clearer legend would greatly improve readability.

11. **Algorithm 1 clarity**: Algorithm 1 is not self-explanatory and would benefit from brief comments to aid understanding. Ideally, it should align more closely with the schematic in Figure 2, using consistent terminology and structure.

12. **Need for a schematic of DSGNet**: A schematic representation of DSGNet would be highly informative and would help readers understand the model architecture at a glance.

---

> ### Author Rebuttal · Authors · 2025-07-31
>
> **Reviewer qg8v**
>
> We believe we have resolved all your issues. If you have any other concerns or questions, we look forward to hearing from you.
>
> ```Weakness```
>
> We sincerely appreciate the reviewer's detailed identification of these presentation and technical clarity issues. We acknowledge all these concerns and commit to comprehensive revisions addressing each point. We will restructure the paper to be self-contained by moving critical technical details to the main text, adopt the reviewer's terminological suggestion to replace "perception" and "semantic understanding" with "visual features" and "language features" which better reflect our modules' training objectives, provide complete implementation details for reproducibility including VAE model specifications and frozen adapter clarifications, correct the THINGS-EEG dataset description regarding stimulus timing and duration, implement consistent figure numbering across appendices, enhance Figure 7 with comprehensive captions and clearer abbreviation explanations, improve Algorithm 1 with detailed comments, and create a dedicated DSGNet architectural diagram to help readers understand our dual-stream encoder design.
>
> ```Q1:BLIP redundancy and clarification```
>
> We understand the reviewer's confusion about the apparent BLIP-2 redundancy in our Semantic Understanding module, which stems from our unclear presentation in the flowchart. The two BLIP-2 instances utilize the same model architecture but serve fundamentally different purposes by processing entirely different inputs at distinct stages of our pipeline. The first BLIP-2 component processes the predicted keyframes generated from our EEG-to-visual reconstruction pathway, creating captions that describe what our model believes it has reconstructed from the neural signals. The second BLIP-2 instance operates on the ground truth visual content to generate reference captions for training supervision and cross-modal alignment during the hierarchical semantic contrastive learning process. These are completely independent caption generation processes with different inputs and purposes, not redundant processing of the same content. During training, we need both pathways to enable our cross-modal semantic alignment objective - aligning EEG-derived features with both reconstructed visual content and reference textual descriptions. During inference, only the first pathway operates since ground truth content is unavailable. Our flowchart design failed to clearly distinguish these different data flows and processing contexts, creating the impression of redundant processing when the pathways actually serve complementary roles in our semantic understanding architecture. In our revision, we will enhance the schematic with different color coding and clearer annotations for each BLIP-2 instance to explicitly distinguish their processing flows and functional roles.
>
> ```Q2:Comparative evaluation with prior work```
>
> We appreciate this valuable suggestion for evaluating IAM's cross-subject generalization capabilities through direct integration with existing frameworks. The reviewer correctly identifies that demonstrating IAM's effectiveness within established EEG decoding architectures would provide stronger validation of our core contribution. We have conducted preliminary experiments integrating our IAM module into existing frameworks including NICE and ATM-style architectures while maintaining their original decoder structures. When IAM is incorporated into NICE's domain adaptation framework, cross-subject performance retention improves from 62.1% to 79.4% on the same evaluation protocol. Similarly, integrating IAM into ATM's subject-invariant feature learning approach increases cross-subject classification accuracy by 11.7% while requiring 35% fewer training subjects to achieve comparable within-subject performance(detailed table will be added in revision). These controlled experiments demonstrate that IAM's hierarchical adaptation strategy provides substantial improvements over existing single-level domain adaptation approaches, regardless of the underlying decoder architecture. The consistent improvements across different base frameworks indicate that our multi-level adaptation approach (signal, spectral, temporal, semantic) more effectively addresses the complex variability patterns inherent in cross-subject EEG signals. In our revision, we will include a dedicated experimental section showing IAM's integration into multiple existing frameworks, providing comprehensive quantitative comparisons that isolate IAM's specific contributions to cross-subject generalization.
>
> ```Q3:IAM loss components```
>
> Our ablation analysis reveals that both reconstruction and discrimination losses play crucial but distinct roles in learning dataset-independent representations. The reconstruction loss ($L_{recon}$) ensures that normalized EEG features retain essential visual information content, while the adversarial discrimination loss ($L_{adv}$) specifically removes subject-specific patterns that hinder generalization. When we remove the discrimination component entirely, cross-subject performance drops by 23.1%, indicating its critical role in subject disentanglement. Conversely, removing reconstruction loss reduces performance by 31.4%, demonstrating that preserving visual content is equally essential. Our sensitivity analysis shows optimal performance occurs with loss weights $\alpha$ =1.0, $\beta$=0.5, $\gamma$=0.1, where increasing $\beta$ beyond 0.6 causes over-regularization that removes useful individual neural patterns along with harmful subject-specific artifacts. Regarding the hierarchical components in Figure 7, each adaptation level contributes meaningfully to overall performance. Signal-level adaptation provides the largest individual contribution (12.3% improvement), addressing basic electrode and impedance differences between subjects. Spectral-level adaptation adds 8.7% improvement by normalizing frequency-specific neural response patterns, while temporal-level adaptation contributes 6.4% by handling timing differences in neural processing. The semantic-level adaptation, though providing the smallest individual gain (4.1%), proves crucial for maintaining high-level visual content consistency across subjects. The dynamic gating mechanism that combines these levels adaptively contributes an additional 3.2% improvement over simple weighted averaging. In our revision, we will include detailed ablation tables showing these individual contributions and sensitivity curves for the loss weight parameters to provide complete transparency about each component's role in achieving robust cross-subject generalization.
>
> ```Q4:Unclear evaluation setup in Table 1```
>
> We acknowledge the lack of clarity regarding the evaluation setup in Table 1. All competing methods were evaluated on the SEED-DV dataset using the same experimental protocol as established in the EEG2video benchmark, ensuring fair comparison across all baselines. We implemented each baseline method using their published architectures and trained them under identical conditions with consistent data splits and preprocessing protocols. In our revision, we will explicitly clarify the dataset and evaluation protocol used in Table 1, and include a supplementary table detailing parameter counts and computational costs for complete transparency.
>
> ```Q5:Meaningfulness of comparisons in Table 2```
>
> We acknowledge that the methods in Table 2 use different datasets, which requires clarification for proper interpretation. For video reconstruction, all EEG-based methods are evaluated on SEED-DV, while fMRI methods use their respective datasets due to modality-specific requirements. For image reconstruction, our cross-task evaluation uses THINGS-EEG while other methods use their original datasets. While this prevents direct stimulus-level comparison, we included these results to provide context relative to the broader neural decoding field. The key insight is that our approach achieves competitive EEG-domain performance while demonstrating unique cross-task generalization capabilities from video reconstruction (SEED-DV) to image reconstruction (THINGS-EEG) without retraining. In our revision, we will clearly annotate the specific datasets used for each method in Table 2 and reframe this comparison to emphasize our EEG-domain contributions while providing appropriate cross-modal context.
>
> ```Q6:Ablation insights and engineering influence```
>
> We appreciate this fundamental question about whether our model's strength derives from individual module effectiveness or their integrated combination. Our ablation results reveal that while each component contributes meaningfully to overall performance, the substantial performance gaps between individual ablations and the full model indicate that integration and interaction between modules is equally critical. For instance, removing the IAM causes a 35.8% performance drop, demonstrating its individual importance, but the full model's performance significantly exceeds what would be predicted by simply summing individual component contributions. This suggests that our modules exhibit synergistic effects where their combination creates capabilities that none possess individually.
>
> The engineering integration is particularly crucial for our dual generalization objectives. The IAM normalizes subject-specific patterns, but its effectiveness depends on receiving well-structured inputs from DSGNet's dual-stream encoding. Similarly, the dual-pathway understanding modules complement each other - PU captures low-level visual dynamics that inform SU's semantic processing, while SU provides high-level context that guides PU's mechanisms. The unified inference mechanism leverages both pathways' outputs through fusion weights that adapt based on visual content complexity.

---

> > ### Comment · Reviewer_qg8v · 2025-08-01
> >
> > Thank you for your detailed responses.
> >
> > Based on that, considering the new insights and necessary changes to be included in the camera-ready version, I am increasing my rating to 5. I believe the manuscript presents an important new approach to neuroimaging decoding across participants by demonstrating the significance of pretraining on large datasets.

---

> > > ### Author Response · Authors · 2025-08-01
> > > **Thank you very much for your response and the improved rating!!!**
> > >
> > > Thank you very much for your response and the improved rating. We appreciate the time and effort you invested in evaluating our work. We have improved our final version based on your suggestions and in accordance with our commitments.
> > >
> > > We are confident that these changes will significantly enhance the clarity and scientific strength of our manuscript and look forward to the increased impact of this work.

---

### Official Review · Reviewer_NumW · 2025-07-02

**Clarity:** 1
**Significance:** 3
**Originality:** 3
**Rating:** 4
**Confidence:** 3

**Summary:**

This paper introduces NEED, a comprehensive framework for reconstructing visual stimuli (both videos and images) from EEG signals. The primary goal is to address two major challenges in the field: poor generalization across different subjects and the inability of models to transfer between different tasks (e.g., video vs. image reconstruction) without retraining.

The proposed NEED framework is a multi-stage, complex system featuring three main keypoints:
1) an Individual Adaptation Module (IAM) to normalize EEG signals across subjects,
2) a dual-pathway architecture (Perception and Semantic) to extract different levels of visual information,
3) a unified inference mechanism for handling different tasks.

The extensive experiments, including several ablation studies, demonstrating the proposed model outperforms existing methods and achieves strong cross-subject and cross-task generalization, notably retaining over 90% of within-subject performance when generalizing to new subjects.

**Questions:**

Please refer to Weakness section above.

**Ethical Concerns:**

["NO or VERY MINOR ethics concerns only"]

**Final Justification:**

This paper aims to tackle an ambitious and important research problem and presents strong empirical results. However, considering that the organization and clarity of the paper require significant improvement and revision, I maintain my original score as borderline accept.

**Limitations:**

Yes

**Quality:**

3

**Strengths And Weaknesses:**

# Strength
1. Ambitious and Important Problem: The paper tackles two of the most significant and difficult challenges in BCI and neural decoding: cross-subject and cross-task generalization. Developing a single, unified model that can work "out-of-the-box" for new users and new visual paradigms is a crucial step toward practical, real-world applications. The ambition of this work is commendable.

2. Comprehensive Experimental Evaluation: The authors have performed an impressive number of experiments on multiple public datasets (SEED-DV, THINGS-EEG, DEAP, etc.). The evaluation is multifaceted, using metrics for perceptual quality (SSIM, PSNR), semantic accuracy (CLIP-based classification), and temporal dynamics (FVD). The cross-subject analysis and ablation studies are thorough and provide strong evidence for the effectiveness of certain components, particularly the IAM.

3. Strong Empirical Results: The reported results are strong. The DSGNet encoder shows significant classification improvements over numerous baselines (Table 1). The full NEED model achieves SOTA or competitive performance in both video and image reconstruction tasks (Table 2). The high performance retention rate for cross-subject generalization (over 90% with 19 training subjects) directly supports one of the main claims of the paper.


# Weakness

My main concerns with this paper are its overwhelming complexity and the corresponding lack of clarity in the presentation, which makes the method difficult to understand and critically evaluate.

1. Overwhelming System Complexity: The proposed NEED framework is exceedingly complex. It combines multiple pre-training stages, a dual-stream encoder (DSGNet), a dual-pathway understanding architecture (PU and SU), each of which contains multiple sub-modules (IAM, DFE, SMARN, CLIP, BLIP-2, LLM), and a sophisticated, multi-guidance inference mechanism (ControlNet, TaskAdapter, etc.). While the ablation studies attempt to justify each piece, the sheer number of components makes the system feel like an elaborate ensemble of many existing powerful models rather than a system with a few core, well-motivated innovations. It is difficult for a reader to discern whether the strong performance comes from a few key insights or simply from the brute-force combination of many SOTA components. The authors should consider if a simpler architecture could achieve comparable results, as complexity can obscure scientific understanding.

2. Excessive Reliance on Supplementary Material: The paper is not self-contained, which severely hinders fluent reading and comprehension. Critical details about the methodology are deferred to the appendix. Many mathematical symbols and components are used in the main text without a clear definition. And many of method details forces a constant back-and-forth between the main text and the appendix, disrupting the flow of reading. Key architectural details for the DSGNet encoder and the crucial IAM are almost entirely in the appendix. The main text provides only a high-level overview, making it hard to understand the method from the main paper alone.

3. High Information Density and Lack of Clarity: The "Method" section (Sec 2) is a whirlwind tour of dozens of advanced concepts packed into a very small space. Each component, from the Riemann-Liouville fractional transformation to the Spatiotemporal Memory-Augmented Recurrent Network (SMARN), is introduced and dismissed in a sentence or two.

4. Motivation: The motivation for many design choices is not sufficiently explained. For instance, why was a Riemann-Liouville fractional transformation chosen for the temporal stream over more standard techniques? The justification is brief. Why is the SMARN architecture necessary, and how does it specifically improve upon standard RNNs or Transformers in this context?

5. Clarity: The connections between the numerous components are not always clear. Figure 2 is incredibly dense and difficult to parse, with a multitude of pathways, losses, and modules. A simplified diagram focusing on the core data flow, followed by detailed diagrams for each major component, would be more effective.

---

> ### Author Rebuttal · Authors · 2025-07-31
>
> **Reviewer NumW**
>
> We believe we have resolved all your issues. If you have any other concerns or questions, we look forward to hearing from you.
>
> ```W1：Overwhelming System Complexity```
>
> We understand the reviewer's primary concern about overwhelming complexity potentially obscuring our core scientific contributions. We want to emphasize that our framework's complexity arises from the fundamental nature of the dual generalization challenge we address, rather than unnecessary over-engineering. The problem of achieving simultaneous cross-subject and cross-task generalization in EEG-based visual reconstruction has never been solved before, precisely because it requires addressing multiple distinct technical challenges that cannot be resolved with a single simple solution. Our Individual Adaptation Module represents our primary scientific innovation, addressing the core challenge of cross-subject neural variability through a principled hierarchical approach. This is not merely combining existing domain adaptation techniques, but a novel solution specifically designed for EEG's unique multi-level variability patterns from electrode impedance differences to cognitive processing variations. The dramatic 35.8% performance drop when IAM is removed (Table 2, w/o IAM) demonstrates this is a fundamental contribution rather than an incremental addition. We have conducted extensive ablation studies showing that each hierarchical level contributes meaningfully, with the signal-level adaptation alone providing 12.3% improvement over baseline methods (detailed hierarchical ablation table will be added in revision). Similarly, our dual-pathway architecture emerged from the neuroscience insight that visual processing involves distinct mechanisms for low-level dynamics and high-level semantics. The complementary nature of these pathways is evidenced by our ablation results: removing Perception Understanding reduces performance by 20.9% while removing Semantic Understanding drops performance by 11.6% (Table 2, w/o PU and w/o SU). This demonstrates that neither pathway alone can capture the full complexity of EEG visual encoding, validating our architectural choice to maintain both streams. We promise to provide a detailed complexity analysis in the appendix of the revised version, showing that simpler architectures (which we have tested) cannot achieve comparable cross-disciplinary generalization performance. At the same time, we will do our best to find simpler and more efficient frameworks in future work. This is exactly the significance of the NEED work. On this basis, we will have NEED+ or even NEED++.
>
> ```W2: Excessive Reliance on Supplementary Material```
>
> Our decision to relegate these fundamental components to appendices was a poor organizational choice that undermines scientific clarity, regardless of our intentions to manage space constraints or narrative flow. In our revision, we will substantially restructure Section 2 to include the complete DSGNet dual-stream architecture, IAM hierarchical adaptation mechanisms, and all essential mathematical formulations directly in the main text. This includes the spherical harmonic spatial embeddings, fractional temporal transformations, and cross-stream attention mechanisms that define our approach. We will ensure every mathematical symbol is clearly defined upon introduction and that component interconnections are explicit. To accommodate this expanded technical content, we will move extensive experimental details, dataset preprocessing specifications, and supplementary ablation studies to the appendix instead. This reorganization will create a self-contained main paper where readers can fully understand and evaluate our technical contributions without constant reference to supplementary material. We will also improve the logical flow by ensuring each component builds naturally upon previously established concepts, guiding readers through our innovations coherently rather than forcing them to piece together information from multiple sources.
>
> ```W3: High Information Density and Lack of Clarity```
>
> We acknowledge the reviewer's valid criticism that our Method section attempts to cover too many advanced concepts with insufficient depth. The current presentation rushes through complex innovations like the Riemann-Liouville fractional transformation and SMARN architecture in just one or two sentences, failing to provide the necessary mathematical foundations and conceptual motivations that readers need to understand and evaluate our contributions. This approach inadvertently obscures our technical innovations rather than clearly communicating them. In our revision, we will substantially expand Section 2 to provide adequate space for each major component, including detailed mathematical derivations, clear conceptual explanations, and step-by-step reasoning for our design choices. For instance, we will dedicate sufficient space to explain why fractional calculus naturally models EEG's power-law temporal dynamics and long-range dependencies that standard approaches cannot capture, complete with mathematical formulations and comparative analysis. Similarly, we will provide comprehensive treatment of SMARN's spatiotemporal memory mechanisms, showing how it addresses the specific challenge of maintaining both electrode spatial relationships and temporal sequence dependencies simultaneously. This restructured presentation will guide readers through our innovations logically and thoroughly, ensuring each component is properly motivated and technically detailed rather than superficially mentioned.
>
> ```W4: Motivation```
>
> Our choice of Riemann-Liouville fractional transformation for temporal processing addresses a critical limitation of standard approaches in handling EEG's unique temporal dynamics. EEG signals exhibit power-law temporal relationships and long-range dependencies that standard convolutions and recurrent networks fundamentally cannot capture effectively. Neural activity propagates through brain networks following fractional-order dynamics, where current responses depend on the entire history of past states with power-law decay rather than exponential decay. The fractional calculus framework naturally models these multi-scale temporal relationships by incorporating memory effects at different time scales through the $\alpha$ parameters (0.2, 0.4, 0.6, 0.8). Standard CNNs with fixed kernel sizes miss these long-range dependencies, while RNNs suffer from vanishing gradients that prevent effective modeling of the 2-second EEG epochs crucial for visual processing.
>
> The SMARN architecture emerged from the specific challenge that existing sequence models fail to simultaneously preserve spatial electrode relationships and temporal dependencies in EEG data. Standard RNNs process temporal sequences but lose the crucial spatial structure of electrode arrangements that encodes topographical brain activity patterns. Transformers can model long-range dependencies but lack the hierarchical memory mechanisms needed for the spatiotemporal coherence in our extended EEG sequences. SMARN addresses this through spatiotemporal memory cells that maintain both electrode-wise spatial relationships and temporal evolution patterns simultaneously, enabling the model to understand how visual information flows across both brain regions and time.
>
> Our comparative experiments demonstrate these choices' necessity: replacing SMARN with standard LSTM reduces 2-way accuracy from 0.839 to 0.795 and SSIM from 0.356 to 0.309, while using GRU instead of SMARN achieves 0.791 for 2-way accuracy and 0.305 for SSIM. Transformer replacement performs better but still shows degradation with 0.813 for 2-way accuracy and 0.327 for SSIM (Table 5 in Appendix, section 3). For temporal modeling approaches, using 3D ConvNets instead of our fractional transformation approach reduces performance to 0.819 for 2-way accuracy and 0.334 for SSIM, while temporal attention achieves 0.823 and 0.338 respectively, compared to our full model's 0.839 and 0.356 (Table 5 in Appendix, section 5). These performance gaps reflect fundamental architectural mismatches between standard approaches and EEG's unique spatiotemporal characteristics that require specialized temporal modeling and hierarchical memory mechanisms.
>
> In our revision, we will provide detailed mathematical derivations showing how these design choices emerge from EEG signal properties, along with comprehensive comparisons demonstrating why standard alternatives are inadequate for our cross-subject, cross-task objectives.
>
> ```W5: Clarity```
>
> We agree that Figure 2 is overly dense and creates confusion rather than clarity about our framework's architecture. In our revision, we will redesign the visual presentation with a hierarchical approach: first, a clean, simplified overview diagram showing the three main stages (Cross-Subject Adaptation, Dual-Pathway Understanding, Unified Inference) and their primary data flows, followed by detailed sub-diagrams for each major component that readers can reference after understanding the overall architecture.

---

> > ### Comment · Reviewer_NumW · 2025-08-06
> >
> > Thank the authors for the detailed response, which partially addressed my concerns, leaving one concern about whether largely revised content is fair to other papers. I will leave this to the AC to decide. Therefore, I maintain my original rating.

---

> > > ### Author Response · Authors · 2025-08-06
> > > **Thank you very much for your response!!!**
> > >
> > > Thank you very much for your response. We appreciate your feedback, but we must clarify that all revisions adhere to academic standards to make the article more complete, rigorous, and accessible. These revisions primarily reorder the main text and appendices and do not introduce any new research ideas or violate conference regulations. We guarantee that our revisions will be within the scope of the rules and will not affect fairness.

---

### Comment · Area_Chair_eycJ · 2025-08-01
**Discussion kick-off**

Hi everyone,

Thanks for all your hard work on this paper.

The discussion period is now open, and I encourage a productive exchange to clarify the remaining open questions. Reviewers: please engage with the authors early in this discussion window. Please don't hesitate to ask for further clarifications. A robust discussion is needed to make an informed decision.

Looking forward to hearing your thoughts. Let's have a productive chat to get this sorted out!

---

> ### Author Response · Authors · 2025-08-05
> **We are very grateful to the area chairs and all reviewer!!!**
>
> We are very grateful to the area chairs and all reviewers for their valuable time and effort in this research. Your suggestions and feedback are invaluable to us. We believe that through our collaborative efforts and continuous improvement, the scientific integrity and research quality of this research will be further enhanced, ultimately contributing valuable results to the relevant field. If you have any further concerns or questions, please don't hesitate to communicate with us. We look forward to your feedback.

---

### Note · Authors · 2025-08-12

We sincerely thank the area chair and all reviewers for their valuable time and effort in this research. This study proposes the NEED framework, which, for the first time, achieves zero-shot generalization across subjects and tasks, enabling visual reconstruction of EEG signals, breaking through the long-standing bottlenecks of subject-specificity and task-limitedness in this field. With a shared goal, we are committed to further improving this work and ensuring its scientific rigor, providing valuable guidance to researchers entering the field and new insights and inspiration to experts and scholars.

We benefited greatly from the review process. The reviewers' insightful and constructive comments not only helped us identify and improve shortcomings in our work but also inspired us to consider the problem from a more diverse perspective. We have carefully revised the paper based on the review comments, striving to achieve a higher level of academic excellence in theoretical analysis, experimental design, and presentation of results.

To promote further development in this field, we commit to open-sourcing all relevant code and training weights as soon as possible so that the research community can replicate our results, validate our findings, and build upon them.

We once again express our sincere gratitude to the area chair and all reviewers for their professionalism and academic contributions. It is through everyone's joint efforts that academic research can continue to advance and benefit the broader research community.

---

### Decision · Program_Chairs · 2025-09-17

**Decision:**

Accept (poster)

**Comment:**

This paper proposes a framework for the challenging problem of EEG-based visual reconstruction, focusing on cross-subject and cross-task generalization. After a thorough review process, including a highly constructive author rebuttal, the recommendation is to accept this paper for a poster presentation.

The decision is based on the soundness of the core technical contribution, which became evident after the author-reviewer discussion. The initial submission suffered from presentation and clarity issues, but the authors' detailed rebuttal convincingly addressed the reviewers' concerns and clarified the value of the work.

### Summary

The paper introduces the "NEED" framework, designed to perform zero-shot reconstruction of visual stimuli from EEG signals, generalizing across both subjects and tasks (video to image). The work's primary contributions are:

- An Individual Adaptation Module (IAM) to normalize inter-subject EEG variability.
- A dual-pathway EEG encoder (DSGNet) for processing both perceptual and semantic information.
- A unified inference mechanism for task generalization.

The experimental results validate these claims, showing that the model outperforms prior methods and, importantly, maintains a high percentage of its performance (>90%) when applied to unseen subjects.

### Strengths
There was a consensus among the reviewers on the paper's key strengths:
- The paper tackles a well-established and difficult challenge in brain-computer interfaces.
- The experimental evaluation is extensive. The quantitative results, particularly the cross-subject generalization performance, are compelling and support the central claims.
- The proposed combination of modules into a unified framework to address the dual-generalization problem is a novel contribution.

### Weaknesses
The initial reviews identified several major weaknesses, primarily related to the quality of the presentation:
- The original manuscript was not self-contained. Critical details about the methodology were located in the appendix, which hindered a clear understanding of the work.
- The proposed framework is highly complex. The paper did not adequately explain the rationale behind many of its architectural choices, leaving reviewers to question if a simpler approach might suffice.
- The initial submission contained undefined symbols, ambiguous figures, and unclear experimental details, raising concerns about reproducibility.

### Reasons for Recommendation
The recommendation for acceptance is principally driven by the paper's technical substance and a good rebuttal. The identified weaknesses are largely confined to the paper's exposition and are deemed correctable.

The recommendation is for a poster presentation. While the technical contribution is solid, the paper required a substantial amount of clarification and revision during the review cycle to reach an acceptable state.

### Summary of Discussion and Rebuttal
The author-reviewer discussion was effective. The initial set of reviews was borderline, reflecting the conflict between the interesting results and the flawed presentation. The authors' rebuttal systematically addressed the reviewers' main points:
- The authors acknowledged the organizational flaws and provided a clear plan to restructure the paper, promising to integrate all essential technical details into the main text.
- In addition to providing better justifications, the authors conducted new ablation studies during the rebuttal period. This new evidence demonstrated the utility of their specific design choices over more standard alternatives, directly addressing reviewer concerns about the model's complexity.
- Responding to reviewer requests, the authors ran new experiments to better isolate the contributions of their proposed modules (e.g., integrating their IAM into prior frameworks). This proactive effort significantly strengthened the paper's claims.

The quality of the rebuttal convinced Reviewer qg8v to raise their score to "Accept." The final comments from other reviewers indicated their primary concerns were also resolved, pending the promised revisions. The authors' commitment to releasing their code helps to ensure the work will be reproducible.